# Comprehensive histochemical profiles of histone modification in male germline cells during meiosis and spermiogenesis: Comparison of young and aged testes in mice

Misako Tatehana[1], Ryuichi Kimura[1], Kentaro Mochizuki[1,2], Hitoshi Inada[1], Noriko Osumi[1]*

1 Department of Developmental Neuroscience, Center for Advanced Research and Translational Medicine (ART), Tohoku University School of Medicine, Sendai, Japan, 2 Department of Medical Genetics, Life Sciences Institute, The University of British Columbia, Vancouver, BC, Canada

* osumi@med.tohoku.ac.jp

**Data Availability Statement:** All relevant data are within the manuscript and its Supporting Information files.

## Abstract

Human epidemiological studies have shown that paternal aging as one of the risk factors for neurodevelopmental disorders, such as autism, in offspring. A recent study has suggested that factors other than *de novo* mutations due to aging can influence the biology of offspring. Here, we focused on epigenetic alterations in sperm that can influence developmental programs in offspring. In this study, we qualitatively and semiquantitatively evaluated histone modification patterns in male germline cells throughout spermatogenesis based on immunostaining of testes taken from young (3 months old) and aged (12 months old) mice. Although localization patterns were not obviously changed between young and aged testes, some histone modification showed differences in their intensity. Among histone modifications that repress gene expression, histone H3 lysine 9 trimethylation (H3K9me3) was decreased in the male germline cells of the aged testis, while H3K27me2/3 was increased. The intensity of H3K27 acetylation (ac), an active mark, was lower/higher depending on the stages in the aged testis. Interestingly, H3K27ac was detected on the putative sex chromosomes of round spermatids, while other chromosomes were occupied by a repressive mark, H3K27me3. Among other histone modifications that activate gene expression, H3K4me2 was drastically decreased in the male germline cells of the aged testis. In contrast, H3K79me3 was increased in M-phase spermatocytes, where it accumulates on the sex chromosomes. Therefore, aging induced alterations in the amount of histone modifications and in the differences of patterns for each modification. Moreover, histone modifications on the sex chromosomes and on other chromosomes seems to be differentially regulated by aging. These findings will help elucidate the epigenetic mechanisms underlying the influence of paternal aging on offspring development.

**Funding:** This work was supported by JSPS KAKENHI Grant Number JP16H06530 to N.O. (https://kaken.nii.ac.jp/ja/grant/KAKENHI-PLANNED-16H06530/) and JSPS KAKENHI Grant Number JP19K18627 to R.K. (https://kaken.nii.ac.jp/ja/grant/KAKENHI-PROJECT-19K18627/). The funders had no role in study design, data collection and analysis, decision to publish, or preparation of the manuscript.

**Competing interests:** The authors have declared that no competing interests exist.

## Introduction

Autism spectrum disorder (ASD) is a neurodevelopmental disorder, and it has been reported to show a pandemic rise in recent decades. The most recent survey in the United States shows that one in 59 eight-year-old children has been diagnosed with ASD [1]. The reason for the rise might include changes in diagnosis, although paternal aging has been suggested as one of the biological factors. For example, meta-analyses of 5,766,794 children born from 1985–2004 in five countries have revealed a significant association between advanced paternal age and ASD risk in children [2].

Aging induces *de novo* mutations in the genome of sperm, and most of the mutations in autistic children are reported to be inherited from fathers [3]. However, a study using a mathematical model has demonstrated that the risk of psychiatric illness from *de novo* mutations derived from advanced paternal age has been estimated as 10–20% [4]. Therefore, some factors other than genetic mutations can influence the health and disease of offspring.

One possible mechanism for paternal aging effects on offspring is changes to the epigenetic information encoded in sperm. Previously, we have reported that sperm derived from aged mice have more hypomethylated regions in their genome that possibly affect offspring's behavior [5]. However, the age-related alteration of histone modifications, another major epigenetic regulation, remains unclear. Sperm is produced through dynamic and complicated processes, termed spermatogenesis, which includes meiosis (in spermatocytes) and morphological changes to produce mature sperm. Considering frequent chromatin remodeling during spermatogenesis, alteration of histone modifications would have a high impact on spermatogenic processes. Additionally, some histone modifications remain in sperm after the histone-to-protamine replacement [6–9]. For example, it has been reported that histones remain in genomic regions that function in early developmental stages [10], and that alteration of histone retention sites in sperm can be inherited by the next generation [11]. Therefore, features of histone moiety in spermatogenesis would provide essential clues for understanding gene regulation after fertilization and subsequent development of offspring. Nevertheless, little is known about the localization patterns of histone modifications in male germline cells and the changes they undergo during aging.

Here, we first produced comprehensive histological profiles of major histone modifications, including histone H3 lysine 9 trimethylation (H3K9me3) and H3K27 acetylation (ac), during murine spermatogenesis, and the individual epigenetic modifications showed dynamic and distinct localization changes. We further examined the influence of aging on histone modifications in each step of spermatogenesis, and we found that the levels of histone modification during spermatogenesis were differentially affected by aging. We also noticed the accumulation of H3K79me3 on sex chromosomes. Further elucidation of how aging-induced epigenetic changes contribute to altered gene regulation in the offspring would provide insight into the influence of paternal aging on the health and disease state of the next generation.

## Materials and methods

### Animals

C57BL/6J mice at 3 months old were purchased from a breeder (Charles River Laboratories, Japan) and were raised to 12 months old at the Institute for Animal Experimentation at the Tohoku University Graduate School of Medicine; they were then used for histological and semiquantitative analyses. All animals were housed in standard cages in a temperature- and humidity-controlled room with a 12-hour light/dark cycle (light on at 8 am), and they had free access to standard food and water. All experimental procedures were approved by the Ethics Committee for Animal Experiments of the Tohoku University Graduate School of Medicine

(#2017- MED210), and the animals were treated according to the National Institutes of Health guidance for the care and use of laboratory animals.

## Histological analyses of histone modifications

Histological analysis was performed based on a previous our report [12]. Mice were anesthetized with isoflurane and perfused with PBS (pH 7.4) to remove blood. After perfusion, the testes were dissected, immersed in 4% PFA in PBS, and cut in half to be postfixed in 4% PFA overnight at 4˚C. After washing in PBS, the testes were cryoprotected by being immersion in 5% and 10% sucrose in PBS solution for 1 hour each and then 20% sucrose in PBS solution overnight; then, the tissues were embedded in OCT compound (Sakura Finetek, Japan). Frozen testis sections were cut at a thickness of 10-μm on a cryostat (Leica, CM3050).

Next, 0.01 M citric acid (pH 6.0) solution was heated using a microwave and kept between 90˚C and 95˚C on a heat plate with gentle stirring. The sections were submerged in the solution for 10 minutes for antigen retrieval. The solution and sections were left to cool to 40–50˚C and were then washed in 1×Tris-based saline containing 0.1% Tween 20 (TBST) (pH 7.4). Then, the sections were blocked by treatment with 3% bovine serum albumin and 0.3% Triton X-100 in PBS. The sections were incubated at 4˚C overnight with the following primary antibodies at a dilution of 1/500: rabbit anti-H3K4me2 (07–030, Millipore), rabbit anti-H3K4me3 (ab8580, Abcam), rabbit anti-H3K9me3 (ab8898, Abcam), rabbit anti-H3K27me2 (ab24684, Abcam), rabbit anti-H3K27me3 (07–449, Millipore), rabbit anti-H3K27ac (ab4729, Abcam), rabbit anti-H3K79me2 (ab3594, Abcam), rabbit anti-H3K79me3 (ab2621, Abcam). Together with the antibody against each histone modification, the antibody mouse anti-SCP3 (ab97672, Abcam), a spermatocyte marker, was applied for easy evaluation for staging the testis tubes. Information on the primary antibodies is summarized in Table 1. Subsequently, either a Cy3- or Alexa488- conjugated secondary antibody (1:500) was applied together with 4'6-diamino-2-phenylindole (DAPI), and the sections were incubated at RT for one hour in a humid chamber. After washing in TBST at room temperature, specimens were mounted in Vectashield (Vector Laboratories, H-1200) or ProLong™ Diamond Antifade Mountant (Invitrogen, P36965). Images were captured using a confocal laser-scanning microscope (LSM780, Carl Zeiss, Jena, Germany).

## Semi-quantitative analyses of histone modifications

Confocal images of testis sections were semiquantitatively analyzed using ImageJ. SCP3 signals were used to select meiotic cells during spermatogenesis. First, we selected the "Region of

**Table 1. List of primary antibodies used in this study.**

| Antibody name | Source | Dilution | Company (Catalogue) | Reference |
|---|---|---|---|---|
| anti-H3K4me2 | rabbit polyclonal | 1/500 | Millipore (07–030) | [13–15] |
| anti-H3K4me3 | rabbit polyclonal | 1/500 | Abcam (ab8580) | [16–18] |
| anti-H3K9me3 | rabbit polyclonal | 1/500 | Abcam (ab8898) | [19–21] |
| anti-H3K27ac | rabbit polyclonal | 1/500 | Abcam (ab4729) | [22–24] |
| anti-H3K27me2 | rabbit polyclonal | 1/500 | Abcam (ab24684) | [25, 26] |
| anti-H3K27me3 | rabbit polyclonal | 1/500 | Millipore (07–449) | [24, 27–29] |
| anti-H3K79me2 | rabbit polyclonal | 1/500 | Abcam (ab3594) | [30–32] |
| anti-H3K79me3 | rabbit polyclonal | 1/500 | Abcam (ab2621) | [33–35] |
| anti-SCP3 | mouse monoclonal | 1/500 | Abcam (ab97672) | [36–38] |
| anti-α-Tubulin (S2 Fig) | mouse monoclonal | 1/500 | Sigma-Aldrich (T6199) | [39, 40] |

All antibodies are commercially available, and their specificity has been reported in previous studies.

Interest (ROI)" by identifying the nuclear area stained with DAPI. Then, the fluorescent intensity of DAPI and histone modifications in the ROI were measured, and the mean intensity was calculated for each signal intensity (total signal intensity/nuclear area). According to a previous literature [41], the mean intensity of histone modifications was subsequently normalized to the mean intensity of the DAPI signal (mean intensity of antibody signal/mean intensity of DAPI signal). We measured the integrated intensity of DAPI signal at each stage to show that DAPI staining was stable and reflects changes of DNA amount (S1 Fig). Compared to the intensity of pachytene spermatocytes, the intensity of round spermatid (after meiosis) was about half. The DNA amount was variable in the preleptotene and metaphase spermatocytes, where DNA synthesis and chromosomal segregation occurs, respectively. Finally, the mean of young group was set as 1, and the intensity of the aged group was plotted as a relative score. We analyzed male germline cells in several tubules from four different young (3 months old) or aged (12 months old) mice, yielding 50 cells to be analyzed for each stage and each histone moiety (50 cells x 12 types of cells were analyzed for each box plot in young group and aged group, respectively). We categorized the normalized intensity of the young testis as–(not detected), + (normalized intensity <0.4), ++ (0.4–0.6), +++ (0.6–0.8), ++++ (0.8–1.0), and +++++ (1.0<), as shown in Table 3. The graphs were drawn using the R [42] and ggplot2 [43] packages. The line on the box indicates the median. Scores of individual cells from young and aged testes are shown as black and red dots, respectively, over box plots in Figs 3-1 to 4-3.

## Statistical analysis

Wilcoxon rank-sum tests were used to determine statistical significance between the young and aged groups at each stage using the statistical analysis software JMP® Pro 14 (SAS Institute Inc., Cary, NC, USA). Values of $^*p < 0.05$ and $^{**}p < 0.01$ were considered statistically significant.

## Results

### 1. Morphological profiles showing histone modification patterns during spermatogenesis

Male germline cells drastically change their morphology during spermatogenesis. Previous studies have shown the localization of H3K79me2/3 in spermatocytes and spermatids as well as the accumulation of H3K4me, H3K9me and H3K27me during transformation from spermatogonia to elongated spermatids [34, 35, 44–47]; however, comprehensive detailed subcellular localization patterns have not been fully examined. Here, we first investigated the localization of major histone modifications from preleptotene spermatocytes to elongated spermatids according to 12 stages based on a previous report [48]. This staging helped us to morphologically follow each step of spermatogenesis without any marker staining.

   (1) Histone modifications that activate gene expression.   First, we examined active marks for gene expression, i.e., H3K4me2, H3K4me3, H3K27ac, H3K79me2 and H3K79me3.

   H3K4me2 signal was observed in all stages of seminiferous tubules. During stage VIII, in which the earliest meiotic spermatocytes (i.e., preleptotene cells) are located near the periphery of the tubule, H3K4me2 was detected in the whole nucleus of preleptotene cells, though the signal was not found in constitutive heterochromatin (Fig 1-1A, arrowhead 1). The signal persisted throughout meiosis (Fig 1-1A, arrowheads 2–6). In the pachytene cells of stage X, a strong signal was detected on the XY body (Fig 1-1A, arrowhead 7). The accumulation of H3K4me2 on the XY body disappeared in the M-phase spermatocytes at stage XII, and only a weak signal was observed in the chromosomes (Fig 1-1A, arrowhead 8). In round spermatids

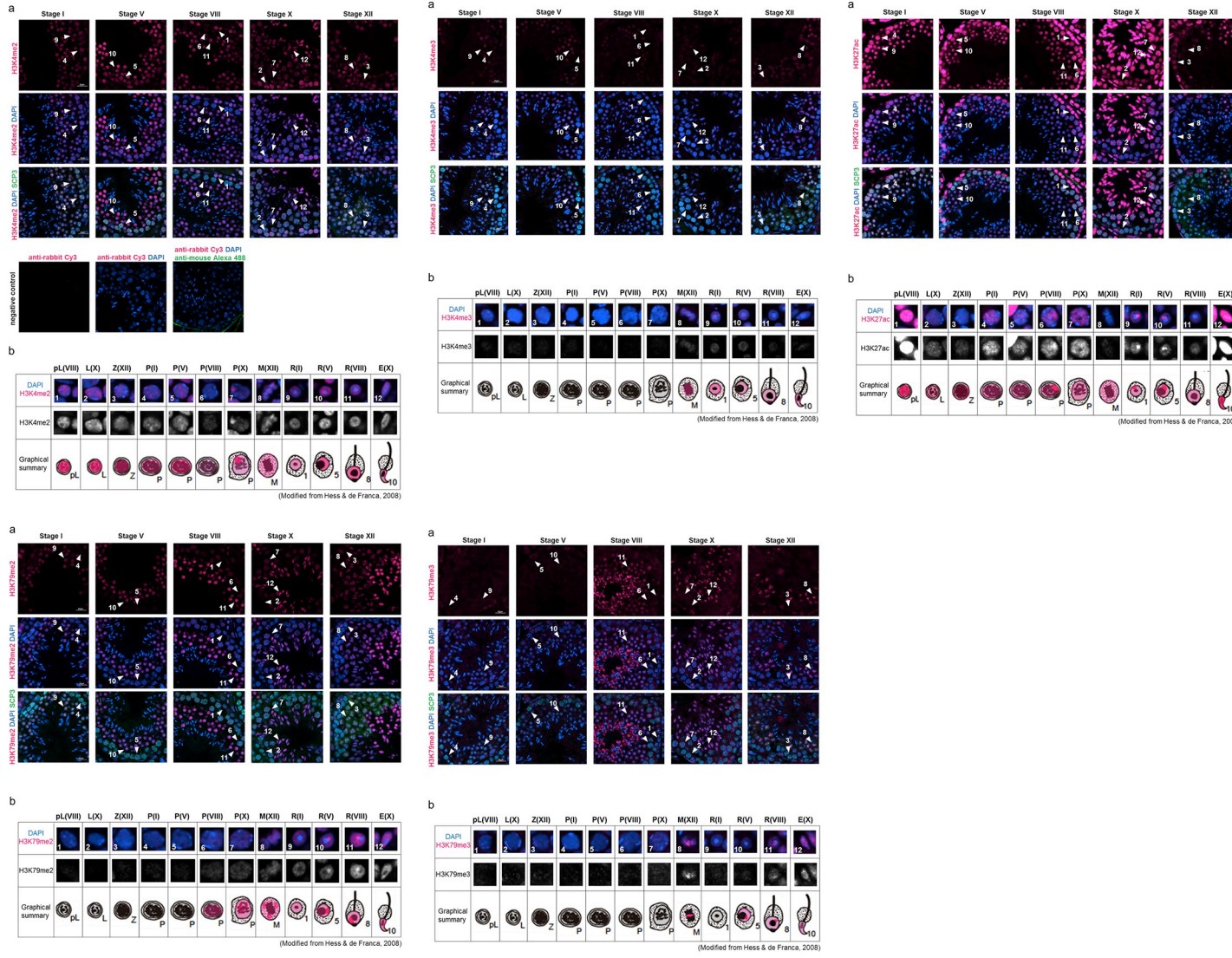

**Fig 1. Localization of H3K4me2 (1–1), H3K4me3 (1–2), H3K27ac (1–3), H3K79me2 (1–4), and H3K79me3 (1–5) in male germline cells in the young (3 M) testis.** (a) Representative confocal images of histone marks (magenta) and SCP3 (green) in the seminiferous epithelium in stages I, V, VIII, X and XII. Nuclei were counterstained with DAPI (blue). The numbers with an arrowhead indicate an individual cell at the specific stage during spermatogenesis, as shown in Table 2. Scale bar: 20 μm. Negative controls (without primary antibodies) are shown in Fig 1–1. (b) Magnified images of the cells indicated by arrowheads in (a), and subcellular localization of each histone modification in the young testis is shown in magenta in a graphical summary. The upper panels are merged images of histone marks (magenta) and DAPI (blue), and the lower panels are grayscale images for the histone marks. Each parenthesis represents the stage of spermatogenesis (see Table 2.).

from stage I to VIII and in elongated spermatids at stage X, the signal was widely detected in the nucleus but not in the chromocenter (constitutive heterochromatin) (Fig 1-1A, arrowheads 9–12). Thus, the H3K4me2 modification seemed to indicate that it was first present in preleptotene cells, and then it accumulated on the XY body in late pachytene cells. Through spermatogenesis, H3K4me2 was not detected in the chromocenter (Fig 1-1B).

H3K27ac signal was observed in all stages of seminiferous tubules. At stage VIII, in which the earliest meiotic cells (i.e., preleptotene cells) were located near the periphery of the tubule, strong H3K27ac signals were detected in whole nuclei of preleptotene cells (Fig 1-3A, arrowhead 1). While the signal continued throughout meiosis, the intensity dramatically decreased when preleptotene cells differentiated into leptotene cells (Fig 1-3A, arrowhead 2–5). In stage

**Table 2. The cells and stages indicated by each number and their abbreviations.**

| Number | Cell and stage | Abbreviation (Stage) |
|:---:|:---:|:---:|
| 1 | preleptotene spermatocyte at stage VIII | pL (VIII) |
| 2 | leptotene spermatocyte at stage X | L (X) |
| 3 | zygotene spermatocyte at stage XII | Z (XII) |
| 4 | pachytene spermatocyte at stage I | P (I) |
| 5 | pachytene spermatocyte at stage V | P (V) |
| 6 | pachytene spermatocyte at stage VIII | P (VIII) |
| 7 | pachytene spermatocyte at stage X | P (X) |
| 8 | M phase spermatocytes at stage XII | M (XII) |
| 9 | round spermatid at stage I | R (I) |
| 10 | round spermatid at stage V | R (V) |
| 11 | round spermatid at stage VIII | R (VIII) |
| 12 | elongated spermatid at stage X | E (X) |

While H3K4me2 was observed beginning in preleptotene cells (Fig 1-1B), the H3K4me3 signal was not detected between preleptotene cells to pachytene cells (Fig 1-2A, arrowheads 1–7). H3K4me3 was first detected in M-phase spermatocytes in stage XII seminiferous tubules (Fig 1-2A, arrowhead 8). The signal was broadly distributed on condensed chromosomes (Fig 1-2B, arrowhead 8). After meiosis II, the signal was detected in round spermatids at stages I to VIII, though there was no signal in the chromocenter (Fig 1-2A, arrowheads 9–11). In elongated spermatids, the signal was weak and widely detected in the whole nucleus (Fig 1-2A, arrowhead 12). Thus, localization of the H3K4me3 modification appeared later in meiosis, and the signal was weak compared to the H3K4me2 modification (Fig 1-2B).

VIII pachytene cells, a strong signal was detected on the XY body (Fig 1-3A, arrowhead 6). The accumulation of H3K27ac on the XY body disappeared in the pachytene cells of stage X and in the M-phase spermatocytes of stage XII, and only a weak signal was observed in the whole nucleus or chromosomes (Fig 1-3A, arrowheads 7–8). In round spermatids from stage I to V, the signal was widely detected in the nucleus, especially in the peri-chromocenter (putative sex chromosome) (Fig 1-3A, arrowheads 9–10). The strong signal at the peri-chromocenter disappeared in stage VIII round spermatids (Fig 1-3A, arrowhead 11). The H3K27ac signal increased again in elongated spermatids, suggesting that H3K27ac is required for spermiogenesis (Fig 1-3A, arrowhead 12). Thus, localization of the H3K27ac modification seemed to indicate that it was first present in preleptotene cells, and then it accumulated on the XY body in late pachytene cells and on the putative sex chromosome during early to middle stages of round spermatids (Fig 1-3B).

In pachytene spermatocytes from stage VIII to stage X, the H3K79me2 signal was widely detected in the nucleus (Fig 1-4A, arrowheads 6 and 7). In M-phase spermatocytes at stage XII, the signal was broadly observed on the condensed chromosomes (Fig 1-4A, arrowhead 8). In round spermatids at stage I, a weak H3K79me2 signal was observed in most of the nucleus, with the exception of the chromocenter (Fig 1-4A, arrowhead 9). Interestingly, a strong H3K79me2 signal was observed in the peri-chromocenter (putative sex chromosome) in round spermatids from stage V to stage VIII (Fig 1-4A, arrowheads 10 and 11), while no signal was observed on the chromocenter itself. In elongated spermatids at stage X, the strong signal on the putative sex chromosome disappeared (Fig 1-4A, arrowhead 12). Thus, localization of the H3K79me2 modification seemed to indicate that it was first present in the late pachytene spermatocytes and to accumulate on the putative sex chromosome during most stages of round spermatids (Fig 1-4B).

While H3K79me2 was observed in late pachytene cells (Fig 1-4B), the H3K79me3 signal was not detected in any pachytene cells. H3K79me3 was first detected in M-phase spermatocytes in stage XII seminiferous tubules (Fig 1-5A, arrowhead 8). Unlike H3K79me2, the

H3K79me3 signal was not distributed on all condensed chromosomes; rather, it was found on specific chromosomes (Fig 1-5A, arrowhead 8). After meiosis II, the signal was not detected in stage I round spermatids (Fig 1-5A, arrowhead 9). However, in round spermatids from stage V to stage VIII, a broad H3K79me3 signal was observed again in the nucleus, and a strong signal was detected in the peri-chromocenter (putative sex chromosome) (Fig 1-5A, arrowheads 10–11). In stage X elongated spermatids, the signal was widely observed in the nucleus (Fig 1-5A, arrowhead 12). Thus, the H3K79me3 modification appeared in the late stage of meiosis and accumulated on the putative sex chromosome of round spermatids. A specific area of condensed chromosomes showed a strong H3K79me3 signal during the M-phase (Fig 1-5B).

**(2) Histone modifications that inhibit gene expression.** Next, we examined repressive marks: H3K9me3, H3K27me2 and H3K27me3.

H3K9me3 was detected in all cell types examined in this study. Broad nuclear signals were observed from preleptotene spermatocytes at stage VIII to pachytene spermatocytes at stage V with strong punctate signals in the periphery of the nucleus (Fig 2-1A, arrowheads 1–5). In pachytene spermatocytes at stage VIII and stage X, only dot-like signals were detected (Fig 2-1A, arrowheads 6–7). These dotted signals matched with intense DAPI signals, indicating areas of heterochromatin. Consistent with a previous report [49], H3K9me3 was especially localized on the sex chromosome of stage XII M-phase spermatocytes (Fig 2-1A, arrowhead 8). In round spermatids of stage I through elongated spermatids of stage X, the signal was detected on the chromocenter and peri-chromocenter (putative sex chromosome) (Fig 2-1A, arrowheads 9–12). Thus, localization of H3K9me3 modification seemed to indicate that it was first present in preleptotene spermatocytes, and then it accumulated on the sex chromosome in M-phase spermatocytes, where it stayed until late round spermatids (Fig 2-1B).

H3K27me2 was detected in the nucleus of stage X leptotene cells through stage X pachytene cells in a salt-and-pepper manner (Fig 2-2A, arrowheads 2–7). In M-phase stage XII spermatocytes, the signal was broadly observed on all chromosomes (Fig 2-2A, arrowhead 8). In round stage I spermatids, strong dot-like signals were observed on the chromocenter, and there were also weak and broad signals in the nucleus (Fig 2-2A, arrowhead 9). The dot-like signals were observed until stage VIII round spermatids (Fig 2-2A, arrowheads 10–11), and they disappeared in elongated spermatids at stage X (Fig 2-2A, arrowhead 12). Thus, localization of the H3K27me2 modification was first observed in leptotene cells, and it accumulated on the chromocenter of all stages of round spermatids; however, the signal was relatively weak through spermatogenesis (Fig 2-2B).

H3K27me3 was detected in all cell types examined in this study except for preleptotene cells. In stage X leptotene cells through stage X pachytene cells, the signal was broadly but faintly detected in the nucleus (Fig 2-3A, arrowheads 2–7). In M-phase stage XII spermatocytes, the signal was widely observed on all chromosomes (Fig 2-3A, arrowhead 8). In stage I round spermatids through stage X elongated spermatids, strong dot-like signals were observed on the chromocenter in addition to the weak and broad signals in the nucleus (Fig 2-3A, arrowhead 12), while strong dot-like signals of H3K27me2 disappeared in elongated spermatids at stage X (Fig 2-3A, arrowhead 12). Thus, localization of the H3K27me3 modification seemed to indicate that it was first present in leptotene cells, which is the same timing as H3K27me2, and then H3K27me3 accumulated on the chromocenter in early round spermatids (Fig 2-3B).

The above localization patterns of histone modifications during spermatogenesis are shown in Table 3. In summary, H3K4me2 was detected in preleptotene cells through elongated spermatids, and H3K27ac was detected in leptotene cells through elongated spermatids. These two modifications were detected at the early stage of spermatogenesis, while other activation marks were detected at the later stage of spermatogenesis. All repressive marks examined in this

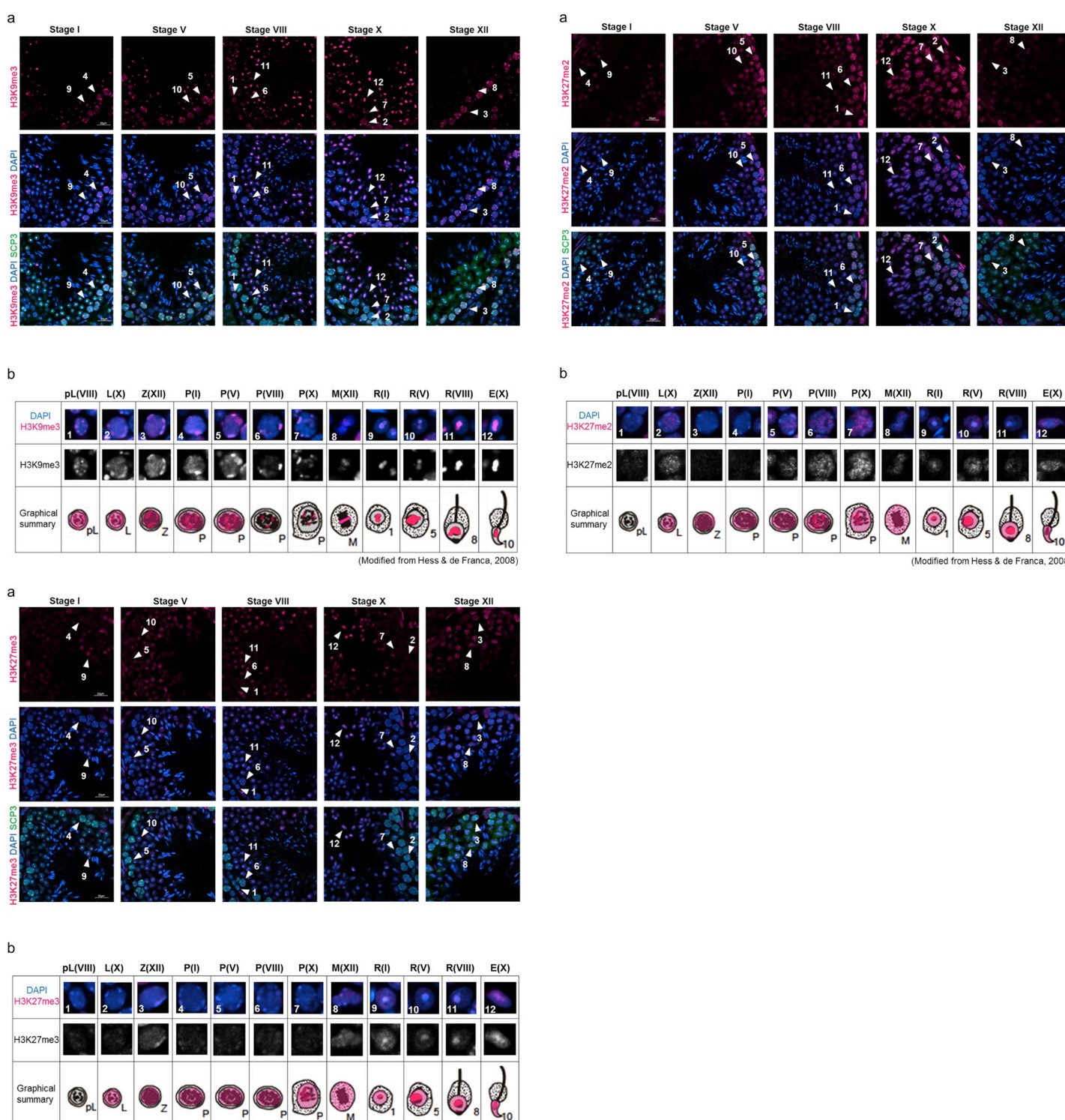

**Fig 2. Localization of H3K9me3 (2–1), H3K27me2 (2–2) and H3K27me3 (2–3) in male germline cells in the young (3 M) testis.** (a) Representative confocal images of histone marks (magenta) and SCP3 (green) in the seminiferous epithelium at stages I, V, VIII, X and XII. Nuclei were counterstained with DAPI (blue). The numbers with an arrowhead indicate individual cells at specific stages during spermatogenesis, as shown in Table 2. Scale bar: 20 µm. (b) Magnified images of the cells indicated by arrowheads in (a), and subcellular localization of each histone modification in the young testis is shown in magenta in a graphical summary. The upper panels are merged images of histone marks (magenta) and DAPI (blue), and the lower panels are grayscale images showing the histone marks. Each parenthesis represents the stage of spermatogenesis (see Table 2.).

**Table 3. Summarized histone modification levels during spermatogenesis and their aging-associates changes.**

| | | pL (VIII) | L (X) | Z(XII) | P(I) | P(V) | P(VIII) | P(X) | M (XII) | R(I) | R(V) | R(VIII) | E(X) | Specific note |
|---|---|---|---|---|---|---|---|---|---|---|---|---|---|---|
| H3K4me2 | Young | ++++ | +++++ | ++++ | +++ | +++ | + | ++ | ++++ | +++++ | +++++ | ++++ | +++++ | Accumulation on XY body |
| | Aged | ↓↓** | ↓↓** | ↓↓** | ↓↓** | ↓↓** | ↓↓** | ↓↓** | ↓↓** | ↓↓** | ↓↓** | ↓↓** | ↓↓** | |
| H3K4me3 | Young | - | - | - | - | - | - | - | ++++ | ++++ | +++++ | ++++ | ++++ | |
| | Aged | - | - | - | - | - | - | - | ↓↓** | → | → | → | → | |
| H3K27ac | Young | +++++ | + | + | + | ++ | ++ | ++ | + | ++ | ++ | ++ | +++++ | Accumulation on XY body and sex chromosomes |
| | Aged | → | ↓↓** | ↓** | * | ↓** | → | ↓↓** | → | ** | → | → | ** | |
| H3K79me2 | Young | - | - | - | - | - | + | + | ++ | ++ | +++++ | +++++ | ++++ | Accumulation on sex chromosomes |
| | Aged | - | - | - | - | - | → | ** | → | → | → | * | ** | |
| H3K79me3 | Young | - | - | - | - | - | - | - | ++++ | | ++ | ++++ | +++++ | Accumulation on sex chromosomes |
| | Aged | - | - | - | - | - | - | - | ** | - | → | ↓* | ↓** | |
| H3K9me3 | Young | ++ | +++ | +++ | +++ | +++ | + | + | + | + | + | ++ | ++ | Accumulation on sex chromosomes and heterochromatin |
| | Aged | ↓↓** | ↓↓** | ↓↓** | ↓↓** | ↓↓** | ↓↓** | ↓↓** | ↓↓** | ↓* | * | ↓* | → | |
| H3K27me2 | Young | - | + | + | + | ++ | ++++ | ++ | ++ | ++ | +++ | ++ | ++ | Accumulation on heterochromatin |
| | Aged | - | ** | ** | ** | ** | ↓* | ** | ** | ** | ** | → | → | |
| H3K27me3 | Young | - | +++ | ++ | ++ | ++ | ++ | ++++ | +++ | +++ | ++++ | +++++ | +++++ | Accumulation on heterochromatin |
| | Aged | - | ↓↓** | ** | ** | ** | ** | ↓↓** | → | ** | ** | → | → | |

Accumulation levels of histone modifications that activate (red) and inhibit (blue) gene expression are shown. Each histone modification level in male germline cells derived from young mice is indicated as–(not detected), + (normalized intensity <0.4), ++ (0.4–0.6), +++ (0.6–0.8), ++++ (0.8–1.0), and +++++ (1.0<) (Actual values for intensity young male germline cells are shown in S1 Table). Arrows show histone modification levels of male germline cells derived from aged mice compared to that of young mice. Single up/down arrows indicate that a signal was increased/decreased by less than 20% of the mean intensity compared to the signal from young animals. Double up/down arrows indicate that the signal was increased/decreased by more than 20% compared to the signal from young animals. Right arrows indicate that there were no significant differences between young and aged animals (*p < 0.05, **p < 0.01). Specific localization or accumulation is described in the specific notes section.

study were detected at the early stages of spermatogenesis. All activation marks except H3K4me3 showed specific accumulation on the XY body in pachytene cells or on a sex chromosome in round spermatids, while all repressive makers accumulated on the chromocenter (i.e., constitutive heterochromatin). Only H3K9me3 accumulated on both sex chromosomes and the chromocenter.

## 2. Semi-quantitative analyses of histone modifications in young and aged testes

We next performed semiquantified analyses to examine whether the intensity of histone modifications is different between young and aged testes. We used 12-month-old mice for the aged group because we previously confirmed that the fertility of >12-month-old male mice was not changed, nor was the litter size of offspring derived from fathers at this age different, yet these offspring showed a significant difference in behavior [5, 50]. Moreover, these results are consistent with data from other papers reporting alterations of social behaviors in adult offspring derived from aged fathers [51, 52]. The histone modification signals were measured by ImageJ. The intensity of histone modifications was normalized to that of DAPI in the nucleus or chromosomes. The intensity of individual cells was plotted to show their individual changes because only one sperm cell will eventually be fertilized to develop a new individual.

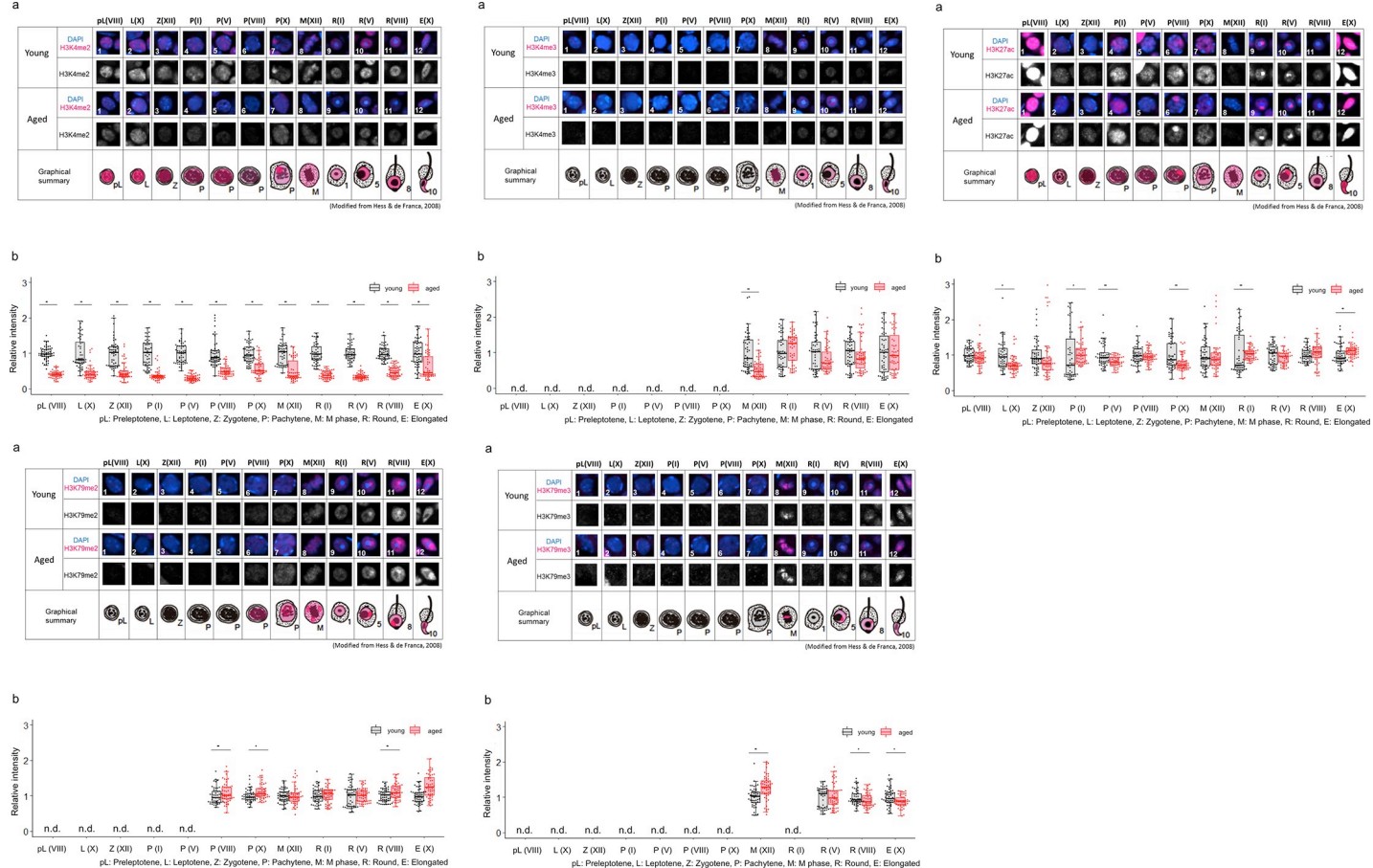

**Fig 3. Comparison of localization patterns and intensity of H3K4me2 (3–1), H3K4me3 (3–2), H3K27ac (3–3), H3K79me2 (3–4) and H3K79me3 (3–5) in male germline cells in young (3M) and aged (12M) testes.** (a) Representative confocal images of histone marks in male germline cells in young and aged testes. In each group, upper panels are merged images of histone marks (magenta) and DAPI (blue), and lower panels show the grayscale images for histone marks. An illustrated overview shows subcellular localization of each histone modification (magenta) in male germline cells in young and aged testes. The numbers indicate individual cells at specific stages during spermatogenesis, as shown in Table 2. (b) Semiquantitative analysis of histone marks in male germline cells in young and aged testes. Gray boxes show the intensity of aged cells relative to young cells (**p < 0.01, *p < 0.05, n.d., not detected). Black and red dots show relative intensity of individual cells from young and aged testes, respectively. Each number in parenthesis represents the stage of spermatogenesis (see Table 2.).

**(1) Histone modifications that activate gene expression.** No difference was observed in the localization patterns of H3K4me2/3, H3K27ac, and H3K79me2/3 in young and aged testes (Fig 3-1A, 3-2A, 3-3A, 3-4A and 3-5A). However, semiquantitative analyses revealed differences in intensity levels among the observed histone modifications.

The intensity levels of H3K4me2 in the cells were lower at all specific stages of spermatogenesis in the aged testis than in the young testis (Fig 3-1B). However, the intensity levels of H3K4me3 were not significantly different in most of the cells. Only in M-phase spermatocytes at stage XII was H3K4me3 intensity lower in the aged testis than in the young testis (Fig 3-2B). The intensity levels of H3K27ac were lower in leptotene spermatocytes at stage X, zygotene spermatocytes at stage XII, pachytene spermatocytes at stages V and X in the aged testis than in the young testis (Fig 3-3B). However, higher intensity was detected in pachytene spermatocytes at stage I, round spermatids at stage I and elongated spermatids at stage X in the aged testis. In other cells, there were no significant differences. Compared with the young testis, higher intensity of H3K79me2 was detected in stage X pachytene spermatocytes, stage VIII round spermatids, and stage X elongated spermatids (Fig 3-4B) of the aged testis. However, other

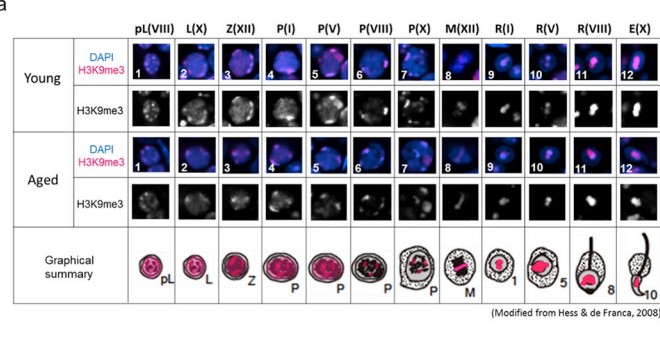

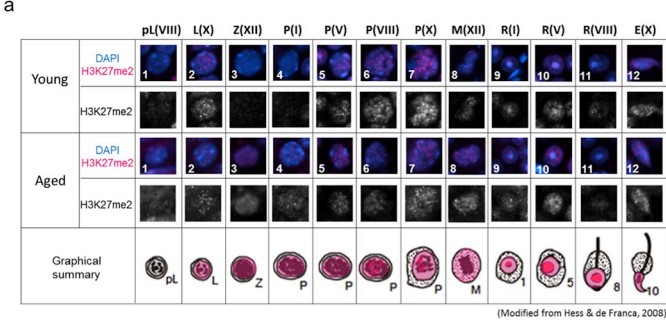

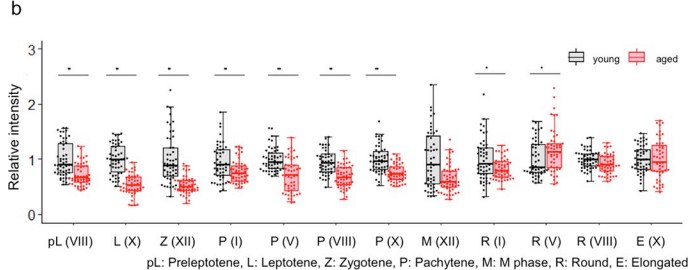

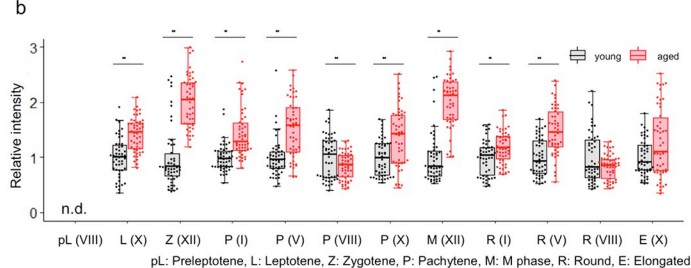

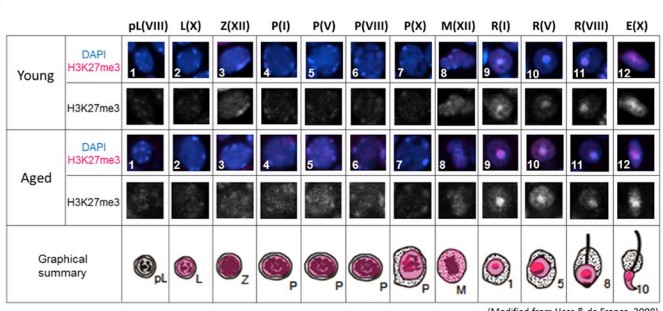

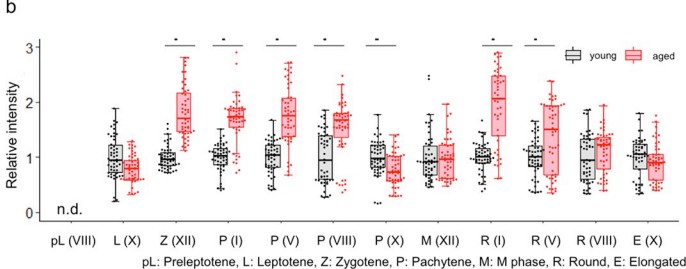

**Fig 4. Comparison of localization patterns and intensity of H3K9me3 (4–1), H3K27me2 (4–2) and H3K27me3 (4–3) in male germline cells in young (3M) and aged (12M) testes.** (a) Representative confocal images of histone marks in male germline cells in young and aged testes. In each group, the upper panels are merged images of histone marks (magenta) and DAPI (blue), and the lower panels show grayscale images for histone marks. An illustrated overview shows subcellular localization of each histone modification (magenta) in male germline cells in young and aged testes. The numbers indicate individual cells at specific stages during spermatogenesis, as shown in Table 2. (b) Semiquantitative analysis of histone marks in male germline cells in young and aged testes. Gray boxes show the intensity of aged cells relative to young cells (**p < 0.01, *p < 0.05, n.d., not detected). Black and red dots show relative intensity of individual cells from young and aged testes, respectively. Each number in parenthesis represents the stage of spermatogenesis (see Table 2.).

cells showed no difference in the intensity level. A higher intensity of H3K79me3 was detected in M-phase spermatocytes at stage XII of the aged testis than in the young testis, while a lower

intensity was observed in round spermatids at stage VIII and elongated spermatids at stage X (Fig 3-5B).

Interestingly, H3K79me3 signal was strongly detected on specific chromosomes in M-phase spermatocytes at stage XII (S2A Fig). According to previous report, H3K79me3 accumulates on sex chromosomes in diakinesis to M-phase spermatocytes [35]. Therefore, we suspected that the chromosome on which H3K79me3 accumulated were sex chromosomes. To identify the sex chromosome that H3K79me3 associated with, we took an advantage of combining immunostaining with fluorescence in situ hybridization (FISH) on the same slide. First, immunostaining was performed to detect H3K79me3 signals in M-phase matocytes (S2B Fig). After images were captured using confocal microscopy, FISH was performed using probes that recognize X and Y chromosomes, which was followed by capturing FISH images of the same slides to obtain three signals (S2C Fig). Signals of H3K79me3 overlapped with those of X and Y chromosomes at the same position within the nucleus of M-phase spermatocytes. Finally, we confirmed that H3K79me3 specifically accumulated on the sex chromosomes in M-phase spermatocytes at stage XII.

**(2) Histone modifications that inhibit gene expression.** No differences were observed in the localization patterns of H3K9me3 and H3K27me2/3 in young and aged testes (Fig 4-1A,4-2A and 4-3A). However, semiquantitative analyses revealed differences in intensity levels among the histone modifications observed.

The intensity levels of H3K9me3 were lower in almost all the cells of the aged testis than it was in the young testis (Fig 4-1B). However, higher intensity was detected in stage V round spermatids in the aged testis. There were no significant differences in elongated spermatids at stage X. In contrast to H3K9me3, the intensity levels of H3K27me2 were higher in almost all cells of the aged testis, and only pachytene cells at stage VIII showed lower intensity in the aged testis than in the young testis (Fig 4-2B). Similar to the results for H3K27me2, almost all stages of cells showed higher intensity of H3K27me3 in the aged testis than in the young testis (Fig 4-3B). Only in stage X leptotene spermatocytes and stage X pachytene spermatocytes were lower intensity detected in the aged testis. There were no significant differences in other cell types.

The above intensity of histone modifications in the young and aged testes is shown in Table 3. In summary, histone modifications were altered by aging, and the alteration patterns were different per the type of each histone modification. Among the repressive marks, H3K9me3 levels were decreased in the aged testis, while H3K27me2 and H3K27me3 levels were increased during spermatogenesis. On the other hand, the intensity level of one activation mark, H3K4me2, was drastically decreased in spermatogenic cells of the aged testis. Other histone modifications also showed differences at some stages.

## Discussion

In this study, we have made a catalog showing the transition of histone modification localization during spermatogenesis and its aging-induced changes. Since our ultimate goal is to understand molecular mechanisms that underlie risks for neurodevelopmental diseases of children from the paternal side, we focused on age-related alteration of histone modification in individual male germline cells. Individual histone modifications showed distinct localization patterns and different timing in terms of the start of their accumulation. Even among the histone modifications that have similar functions, different localization patterns were observed. Moreover, the intensity of histone modifications was altered by aging; aged testes showed different increased/decreased signals of histone modifications at different stages. There were some outliers in certain types of cells and stages; it might be possible that the cells/stages with

several outliers could be critical timings vulnerable to alteration of histone modification during spermatogenesis.

We also identified a chromosome on which H3K79me3 accumulated in M-phase spermatocytes. H3K79me3 accumulated on sex chromosomes in M-phase spermatocytes and round spermatids. Morphologically, it is revealed that not only H3K79me3 but also other histone modifications, such as H3K79me2, H3K27ac and H3K9me3, accumulated on putative sex chromosomes or on the XY body. The intensity of these histone modifications specifically accumulated on the putative sex chromosomes or XY body was also altered by aging at some stages. Therefore, the epigenetic regulation of histone modifications may be different between sex chromosomes and other chromosomes, which may also be affected by aging.

A previous study reported that H3K4me2, a gene expression activation mark, showed dynamic alterations during spermatogenesis. H3K4me2 started to show a strong signal at the preleptotene spermatocytes, became weak in zygotene spermatocytes, and returned stronger again beginning in round spermatids in young mice [45]. Such dynamic alteration of H3K4me2 was also observed in our data; however, the level of H3K4me2 was decreased significantly through spermatogenesis in the aged testis compared to the young testis. It has been reported that overexpression of human KDM1A histone lysine 4 demethylase during spermatogenesis in mice caused a decrease in H3K4me2, leading to various phenotypic abnormalities such as craniofacial and skeletal defects, edema, hemorrhagic gut, extra digits, and missing eyes in E18.5 offspring [53]. The report further shows that the age-related decrease of H3K4me2 may possibly affect the spermatogenic process and that epigenetic alterations can be inherited by the next generation. Considering that H3K4 methylation may play a critical role in spermatogenesis and oogenesis [44, 54], the age-related decrease in H3K4me2 might lead to potential harmful effects in the offspring.

Another example of a histone modification involved in the inhibition of gene expression is H3K9me3. We observed its accumulation in spermatogonia, pachytene spermatocytes, round spermatids and elongated spermatids, and it accumulated on the X chromosome in round spermatids, as reported in previous studies [55–58]. At the pachytene stage of meiosis I, genes on sex chromosomes are transcriptionally shut down by meiotic sex chromosome inactivation (MSCI) and are reactivated after meiosis [59, 60], to which H3K9me3 is reported to be involved [55]. It was also revealed that if there was no enrichment of H3K9me3, remodeling and silencing of sex chromosomes failed and resulted in germ cell apoptosis [49]. In postmeiotic spermatids, 87% of X-linked genes remain suppressed, while autosomes were largely active [61]. Moreover, spermatids form a distinct postmeiotic sex chromatin (PMSC) area that is enriched with H3K9me3 [61, 62]. Thus, H3K9me3, a repressive mark, has various roles during meiosis and in the postmeiotic phase. Our data showed that H3K9me3 appeared first in preleptotene spermatocytes and was observed through elongated spermatids, and then its levels drastically decreased in the aged testis during meiosis and in most of the stages of round spermatids. It is assumed that in the aged testis, failure of correct chromosome segregation during meiosis or leaky expression of X-linked genes might occur in the postmeiotic spermatids, even though the male germline cells safely proceed through meiosis.

Another repressive mark, H3K27me3, was present from leptotene cells to elongated spermatids. A study has reported that H3K27ac may act as an antagonist toward Polycomb-mediated silencing through inhibition of H3K27me3 [63]. H3K27ac signals were observed in all examined stages of male germline cells, especially in preleptotene cells and elongated spermatids. It is very interesting that H3K27ac and H3K27me3 showed exclusive localization in round spermatids; the former accumulated on the sex chromosome, while H3K27me3 accumulated on the chromocenter, a heterochromatic region. Therefore, the absence of H3K27ac (an activation mark) and the presence of H3K27me3 (a repressive mark) in the chromocenter

may be reasonable with regard to gene regulation. We found that H3K27me3 was dramatically increased in the aged testis, while H3K27ac showed a slight decrease at certain stages. H3K27me3 inhibits transcription by inducing a compact chromatin structure [64, 65]. Thus, gene expression influenced by H3K27me3 might be more repressed in the aged testis than in the young testis.

Although there are limitations in our semiquantitative analyses using immunostaining signals, our data showing accumulation of H3K9me3, H3K27ac and H3K79me3 on sex chromosomes are considered to correspond to previous ChIP-seq data [66]. We were able to observe detailed dynamics of histone modifications during spermatogenesis by precisely distinguishing stages and cell types, and such segregation of cell types may be difficult in the preparation of samples for ChIP-seq analyses. A single-cell analysis would thus be interesting to perform in the future.

In a parallel study, we noticed that there were more hypomethylated genome regions in sperm derived from aged mice than there were in sperm from young mice [5]. Here, we observed a drastic decrease of H3K4me2 (an active mark), which seems at first to be contradictory. However, DNA hypomethylation has been observed more frequently in the gene body, not in promoter regions [5]. DNA hypomethylation does not always activate transcription, and methylation in the gene body is reported to be related to transcriptional activation [67]. Therefore, our results seem to be consistent. Moreover, we showed a drastic decrease of H3K4me2 (an active mark) and an increase of H3K27me2/3 (a repressive mark), yet we also noticed a decrease of H3K9me3 (a repressive mark). This seems reasonable because it has gradually been indicated that H3K27me2/3 and H3K9me3 are mutually exclusive [68]. It would be possible that individual histone modifications have different roles and that levels of histone modifications are determined by the balance among chromatin readers, writers and erasers. Therefore, further investigation is needed to determine which genome regions are targets of individual histone modifications and how aging changes these chromatin regulators in male germline cells.

It is not easy to elucidate the function of histone modifications and their changes through aging. However, progeria models can be used to determine aging-induced phenotypes and their underlying mechanisms. Hutchinson-Gilford progeria syndrome (HGPS) patients show premature aging, and cultured cells derived from HGPS patients exhibit increased levels of another repressive mark, H4K20me3, and a reduction of both H3K9me3 and H3K27me3 [69], i.e., histone modifications are inducing regions of heterochromatin [70]. Although this result seems to be rather puzzling based on our general knowledge about parallel regulation of H3K9me3 and H4K20me3, it might be considered that there are fewer heterochromatic regions in HGPS patients. Indeed, loss of heterochromatin is considered to be one of the mechanisms involved in aging [71, 72]. However, H3K27me2/3 was increased at most stages in the aged testis. A similar increase of H3K27me3 due to aging was observed in quiescent muscle stem cells [73]. Therefore, it may be reasonable to assume that the increase/decrease of H3K27me3 may depend on the tissue or cell type. One possible reason for these differences can be explained by the activities of histone methyltransferases or demethylases, some of which have tissue- or cell line-specific functions [74, 75]. Polycomb-repressive complex 2 (PRC2), which catalyzes the methylation of H3K27, transcriptionally represses somatic-specific genes and facilitates homeostasis and differentiation during mammalian spermatogenesis [76]. This specificity can be associated with different alteration patterns of histone modification in each tissue or cell line. Although it is known that PRC2's methyltransferase subunit, EZH1, establishes and maintains H3K27 methylation that is essential for spermatogenesis [77], it is unclear how the accumulation level of H3K27 methylation is increased and affects spermatogenesis in aging. Thus, it might be interesting to investigate the dynamics of histone

modification writers and erasers involved in H3K27 methylation and influence spermatogenesis in aged mice.

The sex chromosomes seemed to be unique in their histone modifications. In this study, we confirmed the accumulation of H3K79me3 on the sex chromosomes in spermatocytes at M phase. A similar localization of H3K9me3 has been shown in a previous study [49]. Moreover, H3K27ac accumulated on the XY body and sex chromosomes in spermatocytes and in round spermatids, respectively. H3K79me2 also showed a signal on the sex chromosomes in round spermatids. We found that the intensity of these histone modifications specifically accumulated on the sex chromosomes and/or XY body were also affected by aging at certain stages. What do those results mean?

According to the database of genes associated with an autism risk (SFARI at June 25, 2019), the X chromosome has 81 risk genes, which is comparable to 95 and 89 risk genes on the larger chromosomes 1 and 2, respectively. It may be reasonable to assume that these three chromosomes have a large number of autism risk genes depending on their size. However, the X chromosome has 16 "syndromic" genes, whereas chromosomes 1 and 2 have only four and five, respectively. Enrichment of the autism risk genes, especially syndromic genes, on the X chromosome might explain more severe symptoms in female patients because the X chromosome of the father is unavoidably transmitted only to a daughter. Therefore, more detailed analyses would be interesting to focus on target genes of histone modifications that are located on the X chromosome.

Our results demonstrated aging-related alterations of histone modifications in spermatogenesis. In the literature, the possibility for the effect of paternal epigenetic factors on offspring's health is suggested [78–82]. Currently, there is no report explaining how the paternal epigenome is actually passed on to offspring. This will be a next important question regarding paternal impact on the physical conditions of offspring. Although most histone proteins are replaced with protamines during spermatogenesis, some histone modifications in sperm remain at promoter regions that are known to regulate embryonic development [10], and altered histone retention sites in sperm can be inherited by the next generation [11]. Further studies are needed to reveal the impact of paternal aging on epigenetic mechanisms that may have a transgenerational impact on the health and disease state of progeny.

## Supporting information

**S1 File. DNA fluorescence *in situ* hybridization (DNA-FISH) followed by immunofluorescence staining.**
(DOCX)

**S1 Table. Mean intensity of histone modification in male germline cells derived from young mice.**
(DOCX)

**S1 Fig. Integrated intensity of DAPI signal in young mice.** Integrated intensity of DAPI signal stained with histone modification signal was measured. Each parenthesis represents the stage of spermatogenesis (see Table 2.). Error bars show S.D.
(TIFF)

**S2 Fig. H3K79me3 localization on sex chromosomes in the spermatocyte at M phase.** (a) An illustrated overview shows subcellular localization of H3K79me3 (magenta). Lower panels show representative confocal images of H3K79me3 (magenta) and α-tubulin (green) in M phase cells (red square in the summarized illustration) in young and aged testes. Nuclei were counterstained with DAPI (blue). Each magnified image of the cells is indicated by the number

shown in the box. (b) Representative confocal images of H3K79me3 (magenta) in M phase cells. Nuclei were counterstained with DAPI (blue). M phase cells are indicated with dotted lines and arrowheads. (c) Representative images of FISH show X (red) and Y (green) chromosomes in M phase cells on the same section as (b). Nuclei were counterstained with Hoechst (blue). The same M phase cells as seen in (b) are indicated with dotted lines and arrowheads. (TIFF)

## Acknowledgments

The authors thank Prof. Yasuhisa Matsui and Dr. Yoshio Wakamatsu for providing insightful comments and assistance, and Ms. Sayaka Makino for animal care. We are also grateful to all members of our laboratory for fruitful discussions and advice.

## Author Contributions

**Conceptualization:** Noriko Osumi.

**Formal analysis:** Misako Tatehana.

**Funding acquisition:** Noriko Osumi.

**Investigation:** Misako Tatehana.

**Methodology:** Misako Tatehana, Ryuichi Kimura, Hitoshi Inada.

**Project administration:** Noriko Osumi.

**Software:** Ryuichi Kimura, Hitoshi Inada.

**Supervision:** Ryuichi Kimura.

**Validation:** Ryuichi Kimura, Kentaro Mochizuki, Noriko Osumi.

**Writing – original draft:** Misako Tatehana.

**Writing – review & editing:** Ryuichi Kimura, Kentaro Mochizuki, Hitoshi Inada, Noriko Osumi.

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
