## [Decision Letter · Decision Letter 0]

1 Oct 2019

PONE-D-19-23798

Detailed profiles of histone modification in male germ line cells of the young and aged mice

PLOS ONE

Dear Dr. Osumi,

Thank you for submitting your manuscript to PLOS ONE. After careful consideration, we feel that it has merit but does not fully meet PLOS ONE’s publication criteria as it currently stands. Therefore, we invite you to submit a revised version of the manuscript that addresses the points raised during the review process.

The reviewers have raised a number of serious questions, especially with the sample size, including proper controls, citing existing knowledge, presenting discussion in a more convincing manner, statistical analyses, presentation of clear and convincing images. Considering the interesting aspect of the results in the manuscript, I am considering this manuscript for major revision, which generally would have been not gone forward with so many concerns raised by the reviewers. Hence, please address all comments carefully.

We would appreciate receiving your revised manuscript by Nov 15 2019 11:59PM. To enhance the reproducibility of your results, we recommend that if applicable you deposit your laboratory protocols in protocols.io, where a protocol can be assigned its own identifier (DOI) such that it can be cited independently in the future. For instructions see: http://journals.plos.org/plosone/s/submission-guidelines#loc-laboratory-protocols

We look forward to receiving your revised manuscript.

Kind regards,

Suresh Yenugu

Academic Editor

PLOS ONE

Journal Requirements:

Reviewers' comments:

Reviewer's Responses to Questions

**Comments to the Author**

1. Is the manuscript technically sound, and do the data support the conclusions?

Reviewer #1: Partly

Reviewer #2: No

Reviewer #3: No

2. Has the statistical analysis been performed appropriately and rigorously? 

Reviewer #1: I Don't Know

Reviewer #2: I Don't Know

Reviewer #3: I Don't Know

3. Have the authors made all data underlying the findings in their manuscript fully available?

Reviewer #1: No

Reviewer #2: No

Reviewer #3: No

4. Is the manuscript presented in an intelligible fashion and written in standard English?

Reviewer #1: Yes

Reviewer #2: Yes

Reviewer #3: No

5. Review Comments to the Author

Reviewer #1: In their manuscript, Tatehana et al present the pattern of expression of several epigenetic marks (histone post translational modifications - PTM) during mouse spermatogenesis by immunofluorescence (IF) on adult testicular sections. They first describe the pattern of expression for each mark in young mice, and then present the differences between young (3-mo old) and old (12-mo old) mice. They conclude that there are some significant differences mainly related to changes in repressive marks and marks associated with the sex chromosomes.

The manuscript is well written (in an intelligible fashion) and the topic of the study (impact of aging on histone PTM in male germ cells) is very interesting but the methodology is not detailed enough to be able to conclude on whether or not there are significant changes.

Major comments:

- The authors describe their observation in young mice but did not cite the many studies which have already described histone PTM pattern of expression during spermatogenesis. Such as (to cite only a few): Van der Heijden et al 2007, Khalil et al 2004, Song et al 2011, Dottermusch et al. 2014,…

If the authors’ findings are novel in some way it would be interesting to talk about the data already published and discuss what novelty the present study adds.

- In general the presentation of the current knowledge (state-of-the-art) in the field is not satisfying. For instance, Line 65, only Hammoud s reference is cited while the literature about the genomic localization of remnant histones is now abundant (and interesting to present because controversial). Another example is the citation about MSCI, which is not the most appropriate (since it is about reactivation of X gene expression after meiosis rather than MSCI process). It would be best to cite a recent review on the topic (such as Meiotic Silencing in Mammals, Turner JM. 2015)

- I did not find information about the number of mice analyzed in each group (young and aged, line 88 “Some mice were raised up to 5- or 12-months old “), nor about the methodology to ‘semi quantify’ IF level by ImageJ. It is therefore not possible to determine if the chosen statistical tests are appropriate. Besides, it would be best if the slides were randomized before analysis.

- The comparison of pattern of expression obtained by IF is interesting and gives valuable qualitative information; yet the ‘quantitative’ comparison has some limits and it has been shown to sometimes produce results that are different from ChIPseq results. These limits should at least be discussed in light of the knowledge of histone PTM genomic localization obtained by ChIPseq analyses (cf. Moretti et al 2016).

Minor comments:

- correct in the abstract the following sentence:

Line 22 “from young (3 months) and aged (12 months) old mice.”

Reviewer #2: General comment

In this study, the authors evaluated in mice the impact of aging on the epigenetic profile of germ cells throughout spermatogenesis. It is a descriptive approach whose originality is to cover all the differentiated cells of the germinal lineage and to compare the adult young males (3 months) to the aged males (12 months).

The authors chose to follow a selection of eight post-transcriptional modifications (PTM) of histones H3 and H4 for their known roles in the regulation of gene expression (repression or activation).

This study is based exclusively on a histological approach (immunofluorescence on testis sections). The data presented are qualitative or semi-quantitative (measurement of fluorescence intensity). The comparison between the two ages reveals no difference in the localization of the labelling inside the germ cells. In contrast, differences in the intensity of the fluorescent labelling (decrease or increase) at different differentiation stages are observed and vary according to the PTM of the histone.

As claimed by the authors, these results shed a light on the aging as a cause of alterations of histones post-transcriptional modifications in the germ cells. However, this conclusion relies on only one experimental approach (immunofluorescence on testis sections) and at least in the actual presentation some of their conclusions are not fully supported by data. In particular, some illustrative image provided are not convincing, and above all, a rigorous description of the experimental procedures and a detailed explanation of the analyses of the data performed are missing.

In this context, major revisions are required to satisfy the main criteria of PLOS ONE concerning experiments (accurate description of methods, sample sizes and statistics).

Major specific comment

1) Technical procedures have to be detailed and clarified. This required a rigorous presentation of the following precisions:

- the number of mice analysed (young and aged) for estimation of the sample size

- the number of testis sections or cells examined for each differentiation stage

- the rational to categorized the staining intensity (to faint, fair, bright, very strong) or how it is related to the mean level of fluorescence intensity.

These all precisions are also required to assess adequacy of the statistic test chosen

2) As the main conclusion of this study relies on the comparison between young and aged mice. The choice of the 12 months-old for aged males deserves to be explained and discuss in regard to the average life span of this mouse strain (28 months) and the criteria defining the efficiency of spermatogenesis (sperm count, fertility, impact on the offspring ...)

3) The presence of the last paragraph of results is questionable. As indicated by the authors the accumulation of the H3K79me3 on sex chromosome have been already published (Ontoso D. et al. in Chromosoma, 2014) and the rational to present a replication of these results relies only on the use of a new protocol combining DNA-FISH and Immunofluorescence. The novelty of this protocol should be clarify by the authors and compare with others previously published (for exemple, Donati C.et al. Journal of Histochemistry &Cytochemistry, 2014).

Minor specific comment

Page 16-17: there is a discrepancy between the text of result and the corresponding Fig. 4 concerning the arrowheads 1 and 2 (pachytene) in the text, but (pre-leptotene/leptotene in the legend and the figure.

Fig.1 to Fig. 11: in all legend of these figure, the number 8 (M phase) is missing.

Fig. 8: the image chosen to illustrated the presence of H3K27me3 at leptotene (L(X) do not allow the visualization of the staining.

Fig.17 a: for Aged illustration the magnified image 1 is not correctly annotated as the two images seem different. Use of the tubulin antibody have to be explained in the legend.

Reviewer #3: This paper profiles post-translational modifications of histone H3 in the male germline of the young and aged testes in mice via immunofluorescence. Although the authors present intriguing results regarding the profiling of these PTMs in spermatogenesis, there are a number of issues to be addressed.

Major

• The authors do not provide enough information regarding the quantitative analysis of imaging data. In particular:

o How many cells were taken for statistical analysis?

o Were the cells taken from one tubule or from several different?

o How many animals were used?

o What do the error bars reflect – standard deviation or the standard error of the mean? How did the authors perform the statistical analysis?

o Which statistical criteria and tests were used?

o An example of how the images were masked in order to quantify signal in individual cell types should also be included to clarify what the signal measurements actually represent.

• There is a certain level of variability in the signal intensity of different antibodies used (especially, comparing fig. 2 and fig. 3) – are these due to the differences in histone PTMs, or because of technical conditions used? The panels of immunofluorescence should also include a negative (no primary antibody) control. It would be advisable to perform a titration experiment for the working concentrations of antibodies, if it was not done.

• It is not entirely clear how the authors did the normalization of the signal. Did they compare the PTM signals to the intensity of DAPI? This is especially crucial for comparing the signals in elongating and elongated spermatids, where histone eviction is taking place. The best option would be to normalize signal intensity to H3 total level, although this is not mandatory.

• Bar plots are clearly not the best way to represent the data variability, since they do not allow to see the outliers, the median value, the quartiles, etc. Box or violin plots would provide a better summary of such data.

• The manuscript should provide all necessary information about the specificity of antibodies used or cite the papers where this has been shown.

• More comprehensive description of PTMs included into the study should be provided, as well as the age-related epigenetic changes in the germline. Please cite more studies in the field, such as Godmann et al. 2007; Jenkins et al. 2015, etc.

Minor

• First, the title of the manuscript doesn’t suggest a specific method of analysis, although the whole study is based on immunofluorescence data. In order not to confuse the readers, it is advisable to make it clear in the title, since the profiling of H3 PTMs can be done by protein- and NGS-based techniques as well.

• The manuscript does not contain the profiling of the PTMs in spermatogonia cells, which represent important stages of spermatogenesis. It is advisable either to include them in the study, or, if not possible, to specify in the title the stages, which were included, so that the scope of the manuscript will be more clear.

• All figures have almost the same description, which is repeated 17 times and occupies a lot of space, even more than the results section.

• Presenting the images in grayscale would make visualization of signal intensity easier to detect by eye. Perhaps using black and white images for individual channels with the merged image in color.

• In the discussion, it would be nice to draw parallels about age-related DNA hypomethylation (Yoshizaki 2019 bioRxiv) and age-related changes in histone PTMs described in this paper, in particular, the decrease of active marks. Can this be related to the expression dynamics of chromatin readers, writers and erasers between the young and the old testes?

• English and typos might be improved.

6. PLOS authors have the option to publish the peer review history of their article (what does this mean?). If published, this will include your full peer review and any attached files.

Reviewer #1: No

Reviewer #2: No

Reviewer #3: No

---

## [Author Response · Author response to Decision Letter 0]

3 Dec 2019

We checked that our manuscript meets PLOS ONE's style requirements.

Point-by-point responses for reviewer’s comments

Reviewer #1: 

In their manuscript, Tatehana et al present the pattern of expression of several epigenetic marks (histone post translational modifications - PTM) during mouse spermatogenesis by immunofluorescence (IF) on adult testicular sections. They first describe the pattern of expression for each mark in young mice, and then present the differences between young (3-mo old) and old (12-mo old) mice. They conclude that there are some significant differences mainly related to changes in repressive marks and marks associated with the sex chromosomes.

The manuscript is well written (in an intelligible fashion) and the topic of the study (impact of aging on histone PTM in male germ cells) is very interesting but the methodology is not detailed enough to be able to conclude on whether or not there are significant changes.

We appreciate that the reviewer highly evaluated our manuscript.

Major comments:

- The authors describe their observation in young mice but did not cite the many studies which have already described histone PTM pattern of expression during spermatogenesis. Such as (to cite only a few): Van der Heijden et al 2007, Khalil et al 2004, Song et al 2011, Dottermusch et al. 2014,…

If the authors’ findings are novel in some way it would be interesting to talk about the data already published and discuss what novelty the present study adds.

As mentioned by Reviewer #1, we agree that our references were insufficient for the discussion of our results. In the revised manuscript, we cited additional studies on histone PTM patterns during spermatogenesis. Our results are consistent with previous findings regarding H3K4me2/3, H3K9me3, and H3K79me in the young testis (p.12, lines 175-177). However, we further observed detailed dynamic patterns of alteration in histone modifications in the aged testis, in which some histone modifications showed changing patterns that were different from those in previous studies. This discrepancy is explained in p.35-36, lines 560-584.

- In general the presentation of the current knowledge (state-of-the-art) in the field is not satisfying. For instance, Line 65, only Hammoud‘s reference is cited while the literature about the genomic localization of remnant histones is now abundant (and interesting to present because controversial). Another example is the citation about MSCI, which is not the most appropriate (since it is about reactivation of X gene expression after meiosis rather than MSCI process). It would be best to cite a recent review on the topic (such as Meiotic Silencing in Mammals, Turner JM. 2015)

We appreciate this comment and have included studies describing histone retention in sperm (Ooi et al.,2007; Chu et al., 2006; Gardiner-Garden et al., 1998; Wykes et al., 2003) in addition to Hammoud et al., (2009); however, histone retention by sperm is still controversial (p.5, lines 64-67). Interestingly, Ben Maamar et al. (2018) mentioned that some of the histone molecules can be retained in sperm and inherited by the next generation (p.5, line 67-68). We also cited other studies (Turner et al., 2015; Khalil et al, 2008) to discuss MSCI (p.32, line 506).

- I did not find information about the number of mice analyzed in each group (young and aged, line 88 “Some mice were raised up to 5- or 12-months old “), nor about the methodology to ‘semi quantify’ IF level by ImageJ. It is therefore not possible to determine if the chosen statistical tests are appropriate. Besides, it would be best if the slides were randomized before analysis.

We revised our “Materials and Methods” section and have added more information, such as the sample size (p.10, lines 141-143). Although it would be ideal to randomize the samples, we prepared all samples simultaneously under the same conditions and analyzed them so that they were not biased.

- The comparison of pattern of expression obtained by IF is interesting and gives valuable qualitative information; yet the ‘quantitative’ comparison has some limits and it has been shown to sometimes produce results that are different from ChIPseq results. These limits should at least be discussed in light of the knowledge of histone PTM genomic localization obtained by ChIPseq analyses (cf. Moretti et al 2016).

As mentioned by Reviewer #1, there are limitations in semiquantitative analyses using immunostaining signals. However, it is simultaneously difficult to perfectly isolate various kinds of male germline cells within the testis and obtain enough amount for quantitative evaluation. For this reason, our approach was to analyze digital images of immunostained testes. Fortunately, we were able to observe detailed dynamics of histone modification during normal spermatogenesis. Additionally, we showed the accumulation of H3K9me3 and H3K27ac on sex chromosomes, which corresponds to the data shown in Moretti et al. (2016). Therefore, we think this is a reasonable approach for profiling and verifying the changes in histone modifications through aging. We added more discussion about this issue in p.33-34, lines 535-541.

Minor comments:

- correct in the abstract the following sentence:

Line 22 “from young (3 months) and aged (12 months) old mice.”

We corrected the sentence to “from young (3 months old) and aged (12 months old) mice.” (p.2, line 23)

Reviewer #2: 

General comment

In this study, the authors evaluated in mice the impact of aging on the epigenetic profile of germ cells throughout spermatogenesis. It is a descriptive approach whose originality is to cover all the differentiated cells of the germinal lineage and to compare the adult young males (3 months) to the aged males (12 months).

The authors chose to follow a selection of eight post-transcriptional modifications (PTM) of histones H3 and H4 for their known roles in the regulation of gene expression (repression or activation).

This study is based exclusively on a histological approach (immunofluorescence on testis sections). The data presented are qualitative or semi-quantitative (measurement of fluorescence intensity). The comparison between the two ages reveals no difference in the localization of the labelling inside the germ cells. In contrast, differences in the intensity of the fluorescent labelling (decrease or increase) at different differentiation stages are observed and vary according to the PTM of the histone.

As claimed by the authors, these results shed a light on the aging as a cause of alterations of histones post-transcriptional modifications in the germ cells. However, this conclusion relies on only one experimental approach (immunofluorescence on testis sections) and at least in the actual presentation some of their conclusions are not fully supported by data. In particular, some illustrative image provided are not convincing, and above all, a rigorous description of the experimental procedures and a detailed explanation of the analyses of the data performed are missing.

In this context, major revisions are required to satisfy the main criteria of PLOS ONE concerning experiments (accurate description of methods, sample sizes and statistics).

We appreciate the comments by Reviewer #2. We revised our summary illustration to show dynamic localization patterns of individual PTMs of histone H3. We also added a detailed description of the experimental procedures with the actual sample size and statistics. Details of each issue are as follows:

Major specific comment

1) Technical procedures have to be detailed and clarified. This required a rigorous presentation of the following precisions:

- the number of mice analysed (young and aged) for estimation of the sample size

- the number of testis sections or cells examined for each differentiation stage

- the rational to categorized the staining intensity (to faint, fair, bright, very strong) or how it is related to the mean level of fluorescence intensity.

These all precisions are also required to assess adequacy of the statistic test chosen

We have revised the “Materials and Methods” and “Results” sections on the following issues (p.10, lines 141-142, and p.10, lines 142-143, p.10, lines 143-145, respectively): 

-The number of mice analyzed: we used four young and aged mice.

-The number of cells analyzed: we semiquantified 30-50 cells for each differentiation stage and each histone moiety.

-The criteria for scores of intensity: we scored the intensity of signals and normalized the data based on the signal in young cells. The scores were calculated as follows: – (not detected), + (normalized intensity <0.4), ++ (0.4-0.6), +++ (0.6-0.8), ++++ (0.8-1.0), or +++++ (1.0<).

2) As the main conclusion of this study relies on the comparison between young and aged mice. The choice of the 12 months-old for aged males deserves to be explained and discuss in regard to the average life span of this mouse strain (28 months) and the criteria defining the efficiency of spermatogenesis (sperm count, fertility, impact on the offspring ...) 

We appreciate this comment. The current study aimed to investigate the influence of paternal aging on the next generation, thus we used male mice at a stage when fertility and litter size were not changed. We have previously examined the altered behavior of pups derived from >12-month-old males (Yoshizaki et al., 2019, bioRxiv; Mai et al., 2019, bioRxiv) and obtained results consistent with those reported in other studies after analyzing the behaviors of adult offspring derived from aged males (Foldi et al., 2010; Janecka et al., 2015). Therefore, we used >12-month-old mice as aged male mice. We added this description in p.23, lines 353-358.

3) The presence of the last paragraph of results is questionable. As indicated by the authors the accumulation of the H3K79me3 on sex chromosome have been already published (Ontoso D. et al. in Chromosoma, 2014) and the rational to present a replication of these results relies only on the use of a new protocol combining DNA-FISH and Immunofluorescence. The novelty of this protocol should be clarify by the authors and compare with others previously published (for exemple, Donati C.et al. Journal of Histochemistry &Cytochemistry, 2014).

As mentioned, Ontoso D. et al. (2014) showed accumulation of H3K79me3 on sex chromosomes during diakinesis to metaphase spermatocytes. They used the SCP3 signal to identify sex chromosomes on spreads of cells. In our study, we used frozen sections to more clearly classify cells into individual stages of spermatogenesis, which makes it rather difficult to identify sex chromosomes using the SCP3 signal. For this reason, we performed DNA-FISH in this study, and we successfully detected the accumulation of H3K79me3 on sex chromosomes of spermatocytes from metaphase to anaphase, which provided novel evidence that has never been previously reported.

Although we searched carefully, we could not find the paper suggested by Reviewer #2 (Donati C. et al. Journal of Histochemistry & Cytochemistry, 2014). Compared to other papers in which Immuno-FISH was used (Yang et al., Chromosoma, 2004; Zinner et al., Advances in Enzyme Regulation, 2007), we developed a novel method by which we performed FISH and immunostaining sequentially on the same section. While in other papers FISH probes are fixed after FISH, we did not do this. We took images after FISH, processed the sections for immunostaining by performing antigen retrieval with boiling citric acid, and took images of the immunostained sections. The images of FISH and immunostaining were then overlaid. We described the details in “Materials and Methods” (p.10-11, lines 150-165).

We consider the significance of our experiment not the novelty of the technique but the identification of sex chromosomes on the frozen section without using SCP3 signals as mentioned above. Our experiment was different from Ontoso et al. (2014) in terms of the stages (diakinesis-metaphase or metaphase-anaphase), material (spread cells or frozen tissues), and the method. With the result of this experiment, we finally succeeded in showing that the increase of H3K79me3 in M phase spermatocytes in the aged testis occurred on the sex chromosomes, in which various key genes for neurodevelopmental diseases exist.

Minor specific comment

Page 16-17: there is a discrepancy between the text of result and the corresponding Fig. 4 concerning the arrowheads 1 and 2 (pachytene) in the text, but (pre-leptotene/leptotene in the legend and the figure.

As mentioned, the number of pachytene cells was incorrect. The signal of H3K79me2 was observed in late pachytene cells, and we thus corrected the arrowheads 1 and 2 to 6 and 7, respectively. Subsequently, other numbers of the arrowheads were also corrected. Moreover, Fig. 1 to 4 were combined together and renamed as Fig. 1-1, Fig. 1-2, and so on, in this revision (p.15-16, lines 232-243).

Fig.1 to Fig. 11: in all legend of these figure, the number 8 (M phase) is missing.

We drastically revised the figure legends to reduce redundancy, and we added Table 2 to indicate each type of cell (including one labeled as 8). (p.17-18, lines 259-271).

Fig. 8: the image chosen to illustrated the presence of H3K27me3 at leptotene (L(X) do not allow the visualization of the staining. 

We changed the image of H3K27me3 in the leptotene stage to a more appropriate image (Fig. 2-3a).

Fig.17 a: for Aged illustration the magnified image 1 is not correctly annotated as the two images seem different. Use of the tubulin antibody have to be explained in the legend.

We changed magnified image 1 in the aged testis (Fig. 5a). The �-Tubulin antibody was described in the legend corresponding to Fig. 5 (p.29, lines 454-464). We also added a list of antibodies (Table 1) on p.9.

Reviewer #3: 

This paper profiles post-translational modifications of histone H3 in the male germline of the young and aged testes in mice via immunofluorescence. Although the authors present intriguing results regarding the profiling of these PTMs in spermatogenesis, there are a number of issues to be addressed.

Major

The authors do not provide enough information regarding the quantitative analysis of imaging data. In particular:

o How many cells were taken for statistical analysis?

Were the cells taken from one tubule or from several different?

How many animals were used?

We used 30-50 cells for each histone PTM in each stage. The cells were chosen from several different tubules per mouse, and we used four mice per young or aged group. We subsequently revised the “Materials and Methods” section (p.10, lines 141-143).

What do the error bars reflect – standard deviation or the standard error of the mean? How did the authors perform the statistical analysis? Which statistical criteria and tests were used?

In the previous version of our manuscript, each of the error bars showed the standard deviation in bar plots. In the revised manuscript, however, we adopted box plots to show the data. We compared the values of each histone modification of various male germ line cells between young and aged testes by Wilcoxon rank-sum tests. Details for statistics are added in Materials and Methods (p.11, lines 168-170).

An example of how the images were masked in order to quantify signal in individual cell types should also be included to clarify what the signal measurements actually represent.

We measured the mean intensity of the signal. Although we did not randomize the samples in the current study, we prepared all the samples simultaneously under the same conditions and analyzed them so that they were not biased.

First, we selected the “Region of Interest (ROI)” by identifying the nuclear area stained with DAPI. Then, the fluorescent intensity of DAPI and histone modifications in the ROI was measured, and the mean intensity was calculated for each signal (total signal intensity/nuclear area). The mean intensity of histone modifications was subsequently normalized to the mean intensity of the DAPI signal (mean intensity of antibody signal/mean intensity of DAPI signal). (p.9-10, lines 134-141)

There is a certain level of variability in the signal intensity of different antibodies used (especially, comparing fig. 2 and fig. 3) – are these due to the differences in histone PTMs, or because of technical conditions used? The panels of immunofluorescence should also include a negative (no primary antibody) control. It would be advisable to perform a titration experiment for the working concentrations of antibodies, if it was not done.

As many researchers are struggling with handling antibodies, we carefully optimized the concentration of the antibodies by introducing negative controls in each experiment (We added negative control images in Fig. 1-1). Some antibodies showed weak signals even at a high concentration, and thus it was difficult to directly compare the level of individual histone modifications. However, the aim of our study was to compare the level of histone modification between young and aged testes. Therefore, we believe that the antibodies used in this study are appropriate for examining alterations in histone modifications during aging.

It is not entirely clear how the authors did the normalization of the signal. Did they compare the PTM signals to the intensity of DAPI? This is especially crucial for comparing the signals in elongating and elongated spermatids, where histone eviction is taking place. The best option would be to normalize signal intensity to H3 total level, although this is not mandatory.

We used DAPI signals to normalize signals from antibodies against histone modifications. The details are described in the “Materials and Methods” of the revised manuscript (p.9-10, line 134-141). We have tried the normalization using panH3 signals. However, the staining of panH3 was not successful because of the antibody we used. 

Bar plots are clearly not the best way to represent the data variability, since they do not allow to see the outliers, the median value, the quartiles, etc. Box or violin plots would provide a better summary of such data.

We agree with this comment, and thus we adopted box plots with outliers using the R and ggplot2 packages in the revised manuscript. We now believe that the data variability is clearly demonstrated. 

The manuscript should provide all necessary information about the specificity of antibodies used or cite the papers where this has been shown.

We used commercial antibodies with specificity that has already been confirmed. We added Table 2 on p.9, which is a list of the antibodies used and references for each. 

More comprehensive description of PTMs included into the study should be provided, as well as the age-related epigenetic changes in the germline. Please cite more studies in the field, such as Godmann et al. 2007; Jenkins et al. 2015, etc.

We appreciate the comment and added more discussion. In our study, H3K4me2, H3K9me3 and H3K27me2/3 showed significant differences during almost all the processes of spermatogenesis, so we added discussion regarding these histone PTMs. We also discussed H3K4me2 in the revised version and included Godmann et al. (2007) (p.31, lines 485-499). Also suggested by Reviewer #2, we included critical papers such as Jenkins et al. (2014) (p.38, lines 605-606). We now believe that the significance of the inheritance of the paternal epigenome is well discussed (p.31, lines 491-496, p.37-38, line 604-614). 

Minor

First, the title of the manuscript doesn’t suggest a specific method of analysis, although the whole study is based on immunofluorescence data. In order not to confuse the readers, it is advisable to make it clear in the title, since the profiling of H3 PTMs can be done by protein- and NGS-based techniques as well.

We agreed with the comment and revised the title as “Comprehensive histochemical profiles of histone modification in male germline cells during meiosis and spermiogenesis: Comparison of young and aged testes in mice”.

The manuscript does not contain the profiling of the PTMs in spermatogonia cells, which represent important stages of spermatogenesis. It is advisable either to include them in the study, or, if not possible, to specify in the title the xstages, which were included, so that the scope of the manuscript will be more clear.

We also have great interest in PTMs in spermatogonia. In the current study, however, we focused on the meiotic stages of spermatocytes because chromatin remodeling is quite active during this period. Technically, we could not find a good marker to simultaneously identify spermatogonial cells and various histone modifications using double staining. In the revised manuscript, we stated that we specified the stages of cells that we analyzed from preleptotene cells to elongated cells (p.12, lines 179-181).

All figures have almost the same description, which is repeated 17 times and occupies a lot of space, even more than the results section.

According to the suggestion by Reviewer #3, Fig. 1 to 5 (inactive marks) were combined into Fig. 1-1 to 1-5, and Fig. 6 to 8 (active marks) were combined into Fig. 2-1 to 2-3 based on the function of histone modifications. Similarly, Fig. 9 to 13 were combined into Fig. 3-1 to 3-5, and Fig. 14 to 16 were combined into Fig. 4-1 to 4-3. This revision of the figures enabled simplification of the corresponding legends. Moreover, explanations of abbreviation and numbers are summarized in Table 2, so redundant sentences were deleted from the figure legends.

Presenting the images in grayscale would make visualization of signal intensity easier to detect by eye. Perhaps using black and white images for individual channels with the merged image in color.

We appreciate this suggestion and added new grayscale images in Figs. 1 through 4. The images show histone modification signals in both young and aged groups to more clearly indicate signal intensity and localization.

In the discussion, it would be nice to draw parallels about age-related DNA hypomethylation (Yoshizaki 2019 bioRxiv) and age-related changes in histone PTMs described in this paper, in particular, the decrease of active marks. Can this be related to the expression dynamics of chromatin readers, writers and erasers between the young and the old testes?

Thank you for the constructive suggestion. We added a description to compare the age-related DNA hypomethylation and histone modifications in the Discussion section on p.34-35, lines 543-558. Previously, we reported that sperm derived from aged mice have more hypomethylated regions in their genome than those of young mice (Yoshizaki et al., bioRxiv, 2019). Here, we showed a drastic decrease in H3K4me2 in aged mice (an active mark). Initially, these results seems to be contradictory. However, DNA hypomethylation was observed more frequently in “gene body” regions, not in “promotor” regions. Since DNA methylation does not always inactivate transcription, methylation in the gene body is reported to be related to transcriptional activation (Rose NR et al., 2014). Therefore, our results seem to be consistent.

Although it is not confirmed in the testis, our methylome analysis in the sperm revealed that KDM1a was hypomethylated near the transcription start site. Since KDM1a demethylates H3K4 and H3K9, the present result are reasonable.

Moreover, we showed a drastic decrease in H3K4me2 (an active mark) and H3K9me3 (a repressive mark) and a drastic increase of H3K27me2/3 (a repressive mark), which also seems to be contradictory. This seems to be reasonable in regard to a recent report that indicated that H3K27me2/3 and H3K9me3 are mutually exclusive (Zhang T et al., 2015). We also think this is because of the difference in their roles. Therefore, further investigation is needed to determine which genome regions are targeted by histone modifications. Furthermore, since the level of histone modifications is determined by the balance among chromatin readers, writers and erasers, it is necessary to verify how these factors change with aging.

English and typos might be improved.

In the revised manuscript, the entire text has been checked by a native English editor.

---

## [Decision Letter · Decision Letter 1]

31 Dec 2019

PONE-D-19-23798R1

Comprehensive histochemical profiles of histone modification in male germline cells during meiosis and spermiogenesis: Comparison of young and aged testes in mice

PLOS ONE

Dear Dr. Osumi,

Thank you for submitting your manuscript to PLOS ONE. After careful consideration, we feel that it has merit but does not fully meet PLOS ONE’s publication criteria as it currently stands. Therefore, we invite you to submit a revised version of the manuscript that addresses the points raised during the review process.

There are still concerns on the number of cells used, the staining procedure and its quantification and statistical analyses.

We would appreciate receiving your revised manuscript by Feb 14 2020 11:59PM. To enhance the reproducibility of your results, we recommend that if applicable you deposit your laboratory protocols in protocols.io, where a protocol can be assigned its own identifier (DOI) such that it can be cited independently in the future. For instructions see: http://journals.plos.org/plosone/s/submission-guidelines#loc-laboratory-protocols

We look forward to receiving your revised manuscript.

Kind regards,

Suresh Yenugu

Academic Editor

PLOS ONE

Reviewers' comments:

Reviewer's Responses to Questions

**Comments to the Author**

1. If the authors have adequately addressed your comments raised in a previous round of review and you feel that this manuscript is now acceptable for publication, you may indicate that here to bypass the “Comments to the Author” section, enter your conflict of interest statement in the “Confidential to Editor” section, and submit your "Accept" recommendation.

Reviewer #1: (No Response)

Reviewer #2: All comments have been addressed

2. Is the manuscript technically sound, and do the data support the conclusions?

Reviewer #1: No

Reviewer #2: Yes

3. Has the statistical analysis been performed appropriately and rigorously? 

Reviewer #1: No

Reviewer #2: Yes

4. Have the authors made all data underlying the findings in their manuscript fully available?

Reviewer #1: Yes

Reviewer #2: Yes

5. Is the manuscript presented in an intelligible fashion and written in standard English?

Reviewer #1: Yes

Reviewer #2: Yes

6. Review Comments to the Author

Reviewer #1: This reviewer appreciate the efforts made by the authors to clarify their experimental procedure. Yet I still have major concerns about it.

1) The authors indicate that “Cells from multiple tubules were measured per testis, and four different young (3 months old) or aged (12 months old) mice were used, yielding 30-50 cells to be analyzed in total.”

which I understand to be 4 young males and 4 aged males: 50 cells in total in the 8 different animals or per animal ? because if it is 50 cells in total, this means, on average 6 cells per animal (per type of cells) which is clearly not enough.

2) I am questioning the quantification protocol. The reasoning behind using DAPI to normalize the antibody signal is unclear to me. Indeed, if there are problems of antibody accessibility or homogeneity within different regions of the slide/section during antibody incubation, they would not be picked up by DAPI staining, which is much more robust than antibody staining (and occurs at the end of the experimental procedure, not in parallel with antibody incubation).

3) Now that data as shown as box plots, one can see outliers as individual dots (sometimes more than 4 per group), and here I am confused because to me the correct way to measure statistical significance is to count (many) cells per individual, calculate the mean intensity, and plot the value obtained per male, and not individual cell values. But according to this figure, this is not how the authors perform their analysis.

Again, the topic of the study (impact of aging on histone PTM in male germ cells) is very interesting but to me the methodology (quantification method, sample size, statistical analyses) is not accurate to be able to conclude on whether or not there are significant changes.

Reviewer #2: In this revised version, the authors provided the important and indispensable informations that were missing previously in paragraph of material and methods. Furthermore, changes introduced in every parts of the manuscript (title, summary, introduction, results, discussion and references) as well as a novel presentation of the experimental data (modification of figures, new table) have considerably improved this new version.

Nevertheless, to satisfy to the PLOS ONE criteria for publication, some modifications have to be introduced again in this revised version. Concerning the semi-quantitative analysis of the fluorescence intensity, explanations given on the experimental procedure, may allow a new presentation for the young males in Figure 1, with the indication of the mean of the fluorescence intensity by stage of differentiation. This would provide and complete the grayscale images of histone marks by a semi-quantitative data. Indeed, these experimental data are not accessible directly in the current presentation, because they are used indirectly through a normalization for the comparison between young and old males in Fig. 3. These data could be added in a modified Figure 1, or at least be accessible as supplementary data.

In their reply, the authors failed to convince (reviewer 2) of the need to maintain a third paragraph of results (H3K79me3 on sex chromosome). As already mentioned, the co-localization of the H3K79me3 mark on XY chromosomes has been documented in metaphase and also suggested in round spermatids in the same publication (Ontoso D. et al., 2014). If the experimental approach developed by the authors (Immuno-FISH) does not justify, according to the authors themselves the novelty of the results presented, these data have to find their place in the supplemental data.

I (reviewer 2) owe you all my apologies for the errors introduced in the reference of publication previously cited for the protocol combining immunofluorescence and DNA-FISH. The right reference is: Donati C. et al, 2004 Journal of Histochemistry & Cytochemistry

PMID: 15385579

DOI: 10.1177/002215540405201009

Finally, some indications for minor modifications are the following:

Page 10

- line 143, 30-50 cells … to be analyzed for each stage and each histone moiety.

- line 146 : complete the citation for R and ggplot2 (check exemple above)

To cite ggplot2 in publications, please use:

- H. Wickham. ggplot2: Elegant Graphics for Data Analysis. Springer-Verlag New York, 2016.

Page 11

line 169: complete the citation for JMP and the following link is suggested

http://www.jmp.com/support/notes/35/282.html

Fig. 3

Line 389: correct the mistake in the legend because there is no arrowhead present in this figure.

7. PLOS authors have the option to publish the peer review history of their article (what does this mean?). If published, this will include your full peer review and any attached files.

Reviewer #1: No

Reviewer #2: No

---

## [Author Response · Author response to Decision Letter 1]

29 Feb 2020

Point-by-point responses for reviewer’s comments

Reviewer #1: 

This reviewer appreciate the efforts made by the authors to clarify their experimental procedure. Yet I still have major concerns about it.

1) The authors indicate that “Cells from multiple tubules were measured per testis, and four different young (3 months old) or aged (12 months old) mice were used, yielding 30-50 cells to be analyzed in total.”

which I understand to be 4 young males and 4 aged males: 50 cells in total in the 8 different animals or per animal ? because if it is 50 cells in total, this means, on average 6 cells per animal (per type of cells) which is clearly not enough.

We apologize that our previous text was misleading. Since a certain type of cells, e.g., leptotene spermatocytes, in the testis were very small in number compared with other types of cells, we measured intensity of additional cells in the testis to obtain the same number of cells to be analyzed. We selected 50 cells for each cell type from four mice to make each box plot. Five to 15 cells (depends on cell types) from 2 or 3 tubules per mice were counted. Finally, we totally analyzed 50 cells for 12 cell types each in a group (for example, 600 cells in young and aged groups, respectively) (p. 10, lines 148-152), and confirmed that our results showed almost the same tendency as in the previous manuscript.

2) I am questioning the quantification protocol. The reasoning behind using DAPI to normalize the antibody signal is unclear to me. Indeed, if there are problems of antibody accessibility or homogeneity within different regions of the slide/section during antibody incubation, they would not be picked up by DAPI staining, which is much more robust than antibody staining (and occurs at the end of the experimental procedure, not in parallel with antibody incubation).

In our immunostaining procedures, all the slides for each histone modification staining were treated simultaneously under the same condition with a great care to ensure that the staining was uniform. Together with the antibody against each histone modification, we used the one against SCP3, a spermatocyte marker, to check the staining was uniform among tubules and to evaluate stages of the testis tubes. Since SCP3 showed robust staining as shown in Figs. 1 and 2, we assessed that our experimental procedures were suitable to obtain stable signals of each histone modification in male germline cells in both young and aged testes. The quality of the histone modification antibodies has been ensured in previous studies (see references listed in Table 1). 

To reduce variable parameter, we used DAPI signal to normalize the histone modification signals in all the stage including elongating and elongated spermatids, which had already been agreed with Reviewer #3. It is difficult to normalize with pan H3 in elongating and elongated spermatids because histone-to-protamine replacement is taking place during these stages.

In this revision, we presented the total DAPI signal intensity at each spermatogenesis stage as supporting information (S2 Fig). Total intensity of DAPI signal dropped half between stage P(X) and R(I), where DNA amount in pachytene spermatocytes becomes half in the round spermatid (after meiosis), indicating that total DAPI signal intensity reflects changes in DNA amount. The DNA amount was variable in the preleptotene and metaphase spermatocytes, where DNA synthesis and chromosomal segregation occurs, respectively. From these results, we thought that the normalization by DAPI signal intensity would be reasonable as a stable standard for normalizing histone modification alteration throughout spermatogenesis (even after protamine replacement). Certainly, this method is semi-quantitative because a net amount of histone and the ratio of histone modification to the total histone is unknown. That is a reason that we had to present histone amount in aged mice as a relative intensity against to the histone amount in young mice. Changes in the net amount of histone during aging are also interesting but it requires further studies in the future. We added the explanation in Materials & Methods (p. 10, line 142-147).

The similar procedures have been applied in other studies (e.g., Ferguson et al., PLoS One, 2013.); the reference is added in the Materials & Methods (p. 10, line 140), and in References. 

3) Now that data as shown as box plots, one can see outliers as individual dots (sometimes more than 4 per group), and here I am confused because to me the correct way to measure statistical significance is to count (many) cells per individual, calculate the mean intensity, and plot the value obtained per male, and not individual cell values. But according to this figure, this is not how the authors perform their analysis.

Again, the topic of the study (impact of aging on histone PTM in male germ cells) is very interesting but to me the methodology (quantification method, sample size, statistical analyses) is not accurate to be able to conclude on whether or not there are significant changes.

We appreciate this encouraging comment from the reviewer, and do hope our re-revision including re-analyses and reforming of data presentation would satisfy him/her.

For the style of data presentation, we did consider the style to show the values per male mice as proposed by this reviewer. We had discussed with the co-authors during our re-revision process, and we have finally adopted the current presentation style, showing scores per cells over the box plots, for the following reasons. First, our ultimate goal is to understand molecular mechanisms that underlie risks for neurodevelopmental diseases of children from the paternal side. We are wondering that altered histone modification in sperm cells could be one of such mechanisms. Since only one sperm cell will eventually be fertilized to develop a new individual, it would not be meaningful to discuss on the mean intensity of the cells per animal; it would mask individual changes in each sperm cell. Actually, the outliers were derived from different male mice. It could be possible that the cells/stages with several outliers could be critical timings vulnerable to alteration of histone modification during spermatogenesis.

Thus, we accordingly modified the text in Materials & Methods (p. 10, lines 147-148), Results (p. 23, lines 354-356), Figure Legends (p. 26, lines 405-406 and p. 28, lines 436-437), and Discussion (p. 30, lines 452-455 and p. 30, lines 460-463).

Reviewer #2: 

In this revised version, the authors provided the important and indispensable informations that were missing previously in paragraph of material and methods. Furthermore, changes introduced in every parts of the manuscript (title, summary, introduction, results, discussion and references) as well as a novel presentation of the experimental data (modification of figures, new table) have considerably improved this new version.

Nevertheless, to satisfy to the PLOS ONE criteria for publication, some modifications have to be introduced again in this revised version. Concerning the semi-quantitative analysis of the fluorescence intensity, explanations given on the experimental procedure, may allow a new presentation for the young males in Figure 1, with the indication of the mean of the fluorescence intensity by stage of differentiation. This would provide and complete the grayscale images of histone marks by a semi-quantitative data. Indeed, these experimental data are not accessible directly in the current presentation, because they are used indirectly through a normalization for the comparison between young and old males in Fig. 3. These data could be added in a modified Figure 1, or at least be accessible as supplementary data.

We appreciate this comment from Reviewer #2. We added S1 Table to show the mean intensity of each type of the cells in the young group as Supporting Information (p. 48, line 849).

In their reply, the authors failed to convince (reviewer 2) of the need to maintain a third paragraph of results (H3K79me3 on sex chromosome). As already mentioned, the co-localization of the H3K79me3 mark on XY chromosomes has been documented in metaphase and also suggested in round spermatids in the same publication (Ontoso D. et al., 2014). If the experimental approach developed by the authors (Immuno-FISH) does not justify, according to the authors themselves the novelty of the results presented, these data have to find their place in the supplemental data.

I (reviewer 2) owe you all my apologies for the errors introduced in the reference of publication previously cited for the protocol combining immunofluorescence and DNA-FISH. The right reference is: Donati C. et al, 2004 Journal of Histochemistry & Cytochemistry

PMID: 15385579

DOI: 10.1177/002215540405201009

We agreed that there is no novelty in our data showing H3K79me3 on the sex chromosomes even though our Immuno-FISH method was newly developed. In the re-revised version, the data were shown as Supporting Information (S3 Fig, p. 25, lines 379-392 and p. 48, line 859-p. 50, line 889).

Finally, some indications for minor modifications are the following:

Page 10

- line 143, 30-50 cells … to be analyzed for each stage and each histone moiety.

We changed the words as Reviewer #2 suggested (p.10, line 150).

- line 146 : complete the citation for R and ggplot2 (check exemple above)

To cite ggplot2 in publications, please use:

- H. Wickham. ggplot2: Elegant Graphics for Data Analysis. Springer-Verlag New York, 2016.

Thank you for suggesting us a suitable reference. We revised the citation for R and ggplot2 by referring the paper suggested (p. 11, lines 154-155).

Page 11

line 169: complete the citation for JMP and the following link is suggested

http://www.jmp.com/support/notes/35/282.html

We checked the link and revised the citation for JMP (p. 11, lines 162-163).

Fig. 3

Line 389: correct the mistake in the legend because there is no arrowhead present in this figure.

We appreciate Reviewer #2’s indication for the mistake that we inadvertently overlooked. We corrected the mistake in the legends of Fig 3 as well as of Fig 4 (p. 26, lines 402-403 and p. 28, lines 433-434, respectively).

---

## [Decision Letter · Decision Letter 2]

12 Mar 2020

Comprehensive histochemical profiles of histone modification in male germline cells during meiosis and spermiogenesis: Comparison of young and aged testes in mice

PONE-D-19-23798R2

Dear Dr. Osumi,

We are pleased to inform you that your manuscript has been judged scientifically suitable for publication and will be formally accepted for publication once it complies with all outstanding technical requirements.

With kind regards,

Suresh Yenugu

Academic Editor

PLOS ONE

Additional Editor Comments (optional):

Reviewers' comments:

Reviewer's Responses to Questions

**Comments to the Author**

1. If the authors have adequately addressed your comments raised in a previous round of review and you feel that this manuscript is now acceptable for publication, you may indicate that here to bypass the “Comments to the Author” section, enter your conflict of interest statement in the “Confidential to Editor” section, and submit your "Accept" recommendation.

Reviewer #1: All comments have been addressed

2. Is the manuscript technically sound, and do the data support the conclusions?

Reviewer #1: (No Response)

3. Has the statistical analysis been performed appropriately and rigorously? 

Reviewer #1: (No Response)

4. Have the authors made all data underlying the findings in their manuscript fully available?

Reviewer #1: (No Response)

5. Is the manuscript presented in an intelligible fashion and written in standard English?

Reviewer #1: (No Response)

6. Review Comments to the Author

Reviewer #1: (No Response)

7. PLOS authors have the option to publish the peer review history of their article (what does this mean?). If published, this will include your full peer review and any attached files.

Reviewer #1: No

---

## [Editor Report · Acceptance letter]

16 Mar 2020

PONE-D-19-23798R2 

Comprehensive histochemical profiles of histone modification in male germline cells during meiosis and spermiogenesis: Comparison of young and aged testes in mice 

Dear Dr. Osumi:

I am pleased to inform you that your manuscript has been deemed suitable for publication in PLOS ONE. Congratulations! Your manuscript is now with our production department. 

With kind regards,

on behalf of

Dr. Suresh Yenugu 

Academic Editor

PLOS ONE